



# Aerial observations of sea ice break-up by ship waves

Dumas-Lefebvre Elie[1] and Dumont Dany[1]

[1]Institut des sciences de la mer de Rimouski, Université du Québec à Rimouski, 310 allée des ursulines, Rimouski, QC, Canada G5L 3A1

**Correspondence:** Elie Dumas-Lefebvre (elie_dumas-lefebvre@uqar.ca)

**Abstract.** We provide the first in situ observations of floe size distributions (FSD) resulting from wave-induced sea ice break-up. In order to obtain such data, an unmanned aerial vehicle was deployed from the Canadian Coast Guard Ship *Amundsen* as it sailed in the vicinity of large ice floes in Baffin Bay and in the St. Lawrence Estuary, Canada. When represented as probability density functions weighted by the surface of ice floes, the FSDs exhibit a strong modal shape which confirms the preferential

size hypothesis debated in the scientific community. Both FSDs are compared to a flexural rigidity length scale, which depends on ice properties, and with the wavelength scale. This comparison tends to show that the maximal distance between cracks is preferentially dictated by sea ice thickness and elasticity rather than by the wavelength. Temporal analysis of one fracture event is also done. Results show that the break-up advances almost as fast as the wave energy and that waves responsible for the break-up propagate following the mass loading dispersion relation. Moreover, our experiments show that thicker ice can

attenuate wave less than thinner ice. This method thus provides key information on the wave-induced FSD, clarifies theoretical aspects from the construction of the FSD to its implementation in models and brings new knowledge regarding the temporal evolution of sea ice break-up.

## 1   Introduction

The marginal ice zone (MIZ) is the ice-covered region that is affected by waves, usually found in the periphery of the polar

and subpolar oceans. Reductions of Arctic sea ice thickness and summer extent in response to global warming (Kwok and Rothrock, 2009; Cavalieri and Parkinson, 2012) generally contribute to increase the extent of the MIZ (Horvat and Tziperman, 2015; Squire, 2020). The decrease of the summer minimum extent at a rate of 10% per decade over the last 30 years (Comiso et al., 2008) provides larger fetch to increasingly frequent cyclones (Rinke et al., 2017) that generate larger waves in the Arctic basin (Smith and Thomson, 2016; Stopa et al., 2016; Li et al., 2019; Thomson and Rogers, 2014; Casas-Prat and Wang, 2020).

These larger waves propagate further and have greater potential to break sea ice up, thus changing the mechanical properties and dynamics of the ice, and of the surrounding ocean and atmosphere.

By fracturing large pieces of sea ice into smaller ones, waves change the floe size distribution (FSD) locally and thus contribute to an increase in the total lateral sea ice surface being in contact with water. This results in a greater total sea ice perimeter and in the exposure to the atmosphere of water areas that were previously capped under a layer of sea ice. During

the melt season, both the increase of the total ice perimeter and the lower albedo caused by the exposure of darker waters





can increase the melt rate (Steele, 1992). As in cold conditions, the enhanced heat loss from the ocean to the atmosphere can promote ice formation.

A fragmented ice cover can also have a significantly different dynamical response to external forces, as discussed by Dumont et al. (2011). Herman et al. (2021) report with great detail, using high resolution satellite imagery, how waves broke up a very large ice floe into much smaller ones, and how easily they drifted and deformed in response to wind, waves and current. Floe size is also important for constraining wave propagation and attenuation. For instance, it determines the flexural response of the ice cover, and consequently the possible scattering (Squire, 2007) of wave energy. Wave scattering by sea ice can be important especially when floe size is comparable or larger than the wavelength (Kohout and Meylan, 2008; Bennetts et al., 2010; Squire, 2018). It also determines the importance of energy dissipation through inelastic and anelastic strain (Boutin et al., 2018).

The FSD is an undoubtedly important parameter for sea ice dynamics, such that there has been great efforts to quantify it. Yet, there are no observations that directly relates the FSD to a given process in the natural environment. Most observations come from satellite or aerial imagery of Arctic and Antarctic MIZs where observable floes have an unknown history (Weeks et al., 1980; Rothrock and Thorndike, 1984; Holt and Martin, 2001; Toyota and Enomoto, 2002; Toyota et al., 2006, 2011; Lu et al., 2008; Herman, 2010; Alberello et al., 2019; Herman et al., 2021). After identifying the boundaries of individual ice floes, either manually or using autonomous image processing algorithms, a characteristic length scale such as the mean caliper diameter (e.g. Rothrock and Thorndike, 1984) is determined and used as a metric for the *floe size*. The FSD is represented either as a number density (ND) or as a probability density function (PDF). The former approach, which is the most widely used, takes the form of a continuous curve relating the number of floes per square kilometres to the floe size in a cumulative or non-cumulative way (see Figure 1 of Stern et al., 2018). In the second approach, floe size categories are given a probability of occurrence based on the frequency of observation in order to represent the FSD as an histogram (see Figure 7 of Herman et al., 2018). It appears from the above-mentioned studies that when represented as a ND, the FSD generally follows a power law of the form $n(D) \propto D^{-\gamma}$ where $n$ is the number of floes having a characteristic floe size $D$. This type of distribution has been tied to the fractal and possibly invariant morphology of fractured sea ice Rothrock and Thorndike (1984), and sometimes presented as a path for understanding the broad characteristics of underlying physical processes (Herman, 2010). A review of the available FSD observations and of the power law is made by Stern et al. (2018). In the end, these observations give a large scale view of the MIZ morphology and can give information on the seasonal evolution of the FSD. However, both the low temporal resolution of satellite images and the sparseness of aerial observations do not allow, for example, to capture individual break-up events. This strongly limits our understanding on how the FSD within a particular MIZ arises from wave-induced break-up, which is crucial for the development of realistic wave-ice interaction models (WIMs).

Large-scale spectral WIMs (Dumont et al., 2011; Williams et al., 2013a, b; Zhang et al., 2016; Bennetts et al., 2017; Boutin et al., 2018; Bateson et al., 2020; Boutin et al., 2020) use a power-law FSD to estimate sea ice morphological properties such as the mean floe size. Such statistical moments are dependent on the shape of the FSD and are further used to parameterize numerous MIZ processes, namely wave-induced break-up. This allows studying floe-size dependent processes on sea ice dynamics, which represents a necessary step for improving global climate models and climate projections. However, by simplifying and idealizing how MIZ processes affect the shape of the FSD, these models do not represent a physically-based solution of the



MIZ dynamics (Herman, 2017). For example, Dumont et al. (2011), Williams et al. (2013a), Williams et al. (2013b), Boutin et al. (2018), Bateson et al. (2020) and Boutin et al. (2020) all model the influence of break-up by updating the maximum floe size ($D_{\max}$) of a power-law FSD even though there is no empirical evidence that this is really how fragmentation affects the shape of the FSD. To our knowledge, Roach et al. (2018) made the first global wave-ice interaction modelling effort that did not

impose a particular form to the FSD. They rather let it evolve according to thermal and mechanical processes similarly to what is proposed by Horvat and Tziperman (2015). By assuming that ice breaks up where the deformation is maximal, Roach et al. (2018) obtained that the fracture of sea ice by waves leads to a preferential size. This means that wave-induced break-up leads to a bell-shaped FSD, a result that indicates that the morphology of floes resulting from breakup might not be well represented by a power law.

Focusing on the break-up itself rather than on its influence on dynamics, Fox and Squire (1991) studied the propagation of strain into an ice sheet. Modelling sea ice as a thin, semi-infinite elastic plate, they obtained that "*the position [of maximum strain] depends crucially on ice thickness and to a lesser extent on wave period*". On the other hand, Herman (2017), by modelling the ice cover as a strip of discrete cubic-shaped grains being linked by elastic bonds, obtained that "*breaking of a continuous ice sheet by waves produces floes of almost equal size, dependent on thickness and strength of the ice but not on the*

*characteristics of the incoming waves*". Hence, despite using different frameworks and assumptions, Herman (2017) and Fox and Squire (1991) come to the same conclusions that i) floe size resulting from wave-induced break-up does not depend on the spectral structure of ocean waves, but rather on sea ice thickness and rigidity, and that ii) a preferential size is created by this mechanism. However, the question remains as to whether these theoretical conclusions are supported by field observations.

Although there has been significant efforts to model the break-up process, only few studies have approached the problem

from an observational perspective (e.g. Langhorne et al., 1998; Kohout et al., 2014). Furthermore, little attention has been put towards the analysis of the resulting FSD and its possible connection to sea ice thickness and rigidity. The first anecdotal observations of wave-induced breakup are recalled in a review of the wave-ice interaction topic by Squire (1995). Therein, he states that "*the width of the strips [...] created by the process is remarkably consistent and appears [...] to be rather insensitive to the spectral structure of the sea but highly dependent on ice thickness*". In other words, he observed that the distance between

successive cracks generated by break-up seems to be constant and independent from the sea state but rather dependent on the material properties of sea ice. This remark qualitatively supports the conclusions of Fox and Squire (1991) and Herman (2017), but quantitative analysis is required to fully test these hypotheses. More recently, Herman et al. (2018) carried out break-up experiments in large tanks where waves were generated artificially to break apart a layer of laboratory-grown ice with the goal of testing the conclusions of Herman (2017) and Squire (1995) on the independence of the break-up pattern on wave properties.

Herman et al. (2018) compared the mean sizes obtained from the experiments to a theoretical fracture distance $x^*$ derived by Mellor (1983), which is dependent on the flexural rigidity of the material considered, in this case ice, but not on wave period or length. For a group of experiment (group A test 2060), the value of $x^*$ was close to the mean size so that Herman et al. (2018) concludes that "*the floe size resulting from breaking by waves depends not on the incoming wavelength, but rather on the mechanical properties of the ice itself*". In other words, it means that waves produce floes that are anisotropic and that

the minor axis of these floes is independent of the wavelength. Unfortunately, a factor of $\frac{1}{2}$ was omitted in the mathematical





expression of $x^*$ (as demonstrated in section. 5.1) so that this conclusion needs to be revisited. With that in mind, observations of sea ice break-up by waves in the natural environment are needed to test the flexural rigidity-dependent preferential size hypothesis and to better understand the overall FSD generated by this process. Determining such details from observations would help to develop physically-based parameterizations of wave-induced break up in WIMs.

Few studies about break-up have been made yet mainly because the MIZ is an arduous area to sample directly from. It is indeed hard to be in the MIZ at the right place and at the right time, with good but not too harsh weather conditions for break-up to happen, and with the right apparatus to measure all relevant variables during a natural break-up event. Indeed it is possible to study wave-ice interactions in laboratories as Herman et al. (2018) did, but it is not clear if the results directly apply to the natural environment due to the difference of scale and properties between laboratory-grown ice and sea ice. In this study, we
use a ship to generate waves and an unmanned aerial vehicle (UAV or drone) to film the break-up of a large sea ice floe in the natural environment. Using image processing techniques, we are able to characterize the evolution of the break-up and the resulting FSD. This paper reports on two experiments conducted in the Gulf of St. Lawrence (GSL) and in Northern Baffin Bay (NBB).

## 2    Methods

The setup for the experiments conducted to obtain FSDs resulting from wave-induced breakup is as follows. First, a large level ice floe having a side exposed to open water is localized. Then a UAV is deployed and positioned above the ice edge to record high resolution footage of the break-up event. Finally, the Canadian Coast Guard Ship (CCGS) *Amundsen* cruises near the floe edge at a high and constant speed such that waves are generated in the vicinity of the ice. Two experiments are carried out, the first one in February 2019 in the GSL and the second in August 2019 in NBB. In both experiments, no wind-generated
wave nor swell were present, hence the observed break-up can only be attributed to ship-generated waves. The use of a ship for these experiments hands a better management of weather conditions for drone deployments while still allowing to study break-up in the natural environment. Such a setup also allows to have no constraint on the location of deployments and to search for the right sea ice to break. A DJI Mavic 2 Pro is used for both experiments because of its autonomy, resilience to cold temperature, hovering stability and high resolution camera. The camera has a 65.5° field of view and a sensor that record
photos at a resolution of 20 Mpx and 4K videos. The camera is factory-calibrated by DJI. While its height relative to the takeoff location is obtained by a barometric sensor, it uses both GPS and GLONASS for geopositioning, hovering in a still position and correcting its altitude. The error on its vertical and horizontal position are respectively of 0.5 and 1.5 m. The specifics of each experiment are described below.

### 2.1    Gulf of St. Lawrence

The first experiment was conducted in the northwestern Gulf of St. Lawrence (GSL), Canada, at (49.584°N, 66.152°W) on 7 February 2019 during clear sky conditions and air temperature of $T_{air} = -7.2°C$. According to the ice chart produced by the Canadian Ice Service (CIS) on the same date, there was 9+/10 of grey and grey-white ice, between 10 and 30 cm-thick.



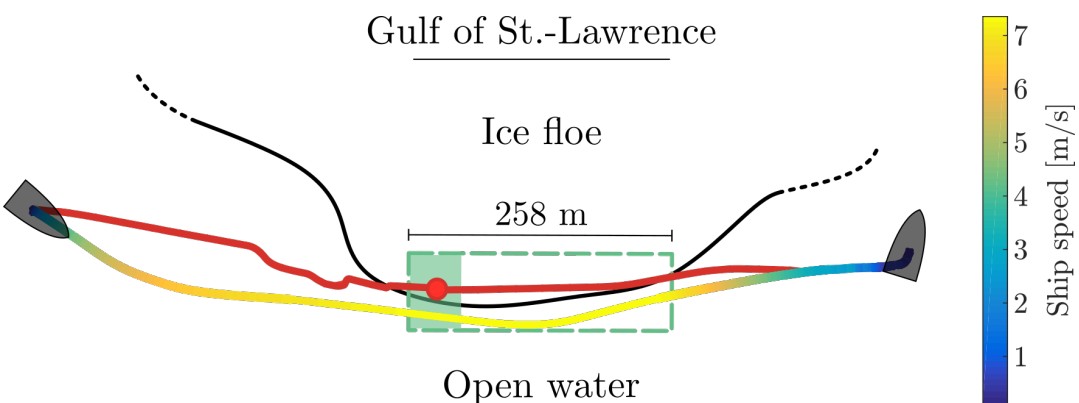

**Figure 1.** Schematic representation of the experiment conducted in the GSL. The grey shape represents the ship. The red and blue-to-yellow lines are respectively the UAV and ship GPS tracks. The red dot is the position where the drone filmed the break-up from a fixed position, and the filled green rectangle is its field of view. The dashed green rectangle is the area covered by the panoramic picture obtained by taking multiple pictures of the resulting broken ice (Figure 7). The black line shows the approximate location of the floe edge.

Figure 1 shows the schematic of the GSL experiment. The *CCGS Amundsen* accelerated and reached an average speed of $7.25 \text{ m s}^{-1}$ less than 50 m from the region of interest, thereby generating a wave train with an amplitude that was sufficient to

break up the ice floe. The ship speed over the portion of the trajectory that generated the observed waves remained relatively uniform, with a standard deviation of $0.08 \text{ m s}^{-1}$. Oblique sunlight conditions allowed us quantifying the wavelength, the period and the direction of the waves (see. section 3.2). Unfortunately, we could not deploy any wave sensor on the ice or in the water so that we can't quantify wave amplitude during the experiment. Using UAV's mean altitude ($94.7 \pm 0.5 \text{ m}$) and the camera field of view, a metric conversion factor of $3.1 \text{ cm px}^{-1}$ is obtained with an uncertainty of ca. $0.01 \text{ cm px}^{-1}$ mostly

due to altitude variations.

### 2.2   Northern Baffin Bay

The second experiment was conducted in northern Baffin Bay (NBB) at (77.883°N, 77.341°W) on August 5 2019 in cloudy conditions and with air temperature of $T_{\text{air}} = 4.9°C$. According to the ice chart produced by the Canadian Ice Service (CIS) on this day, ice concentration was 4/10 and the ice thickness was identified to be between 30 and over 120 cm as thin and thick first year ice were indicated to be present. The floe chosen for the experiment consisted in a plate of heavily rotten first year ice

of about 540-m wide and more than 2-km long. The thickness was assessed on some floes resulting from the experiment using a meter stick and was measured to be between 40 and 60 cm.

Figure 2 describes the experimental setup in NBB. The average speed of *CCGS Amundsen* evaluated during its passage near the floe is $8.37 \text{ m s}^{-1}$ with a standard deviation of $0.05 \text{ m s}^{-1}$. In this experiment, an attempt was made to deploy wave buoys

on the ice with the zodiac before waves hit and fractured the floe. Two SKIb wave buoys (Guimarães et al., 2018) were installed on the floe 2 to 3 m of the edge and roughly 10 m apart in order to measure the wavelength. The zodiac then went at a safe



# Northern Baffin Bay

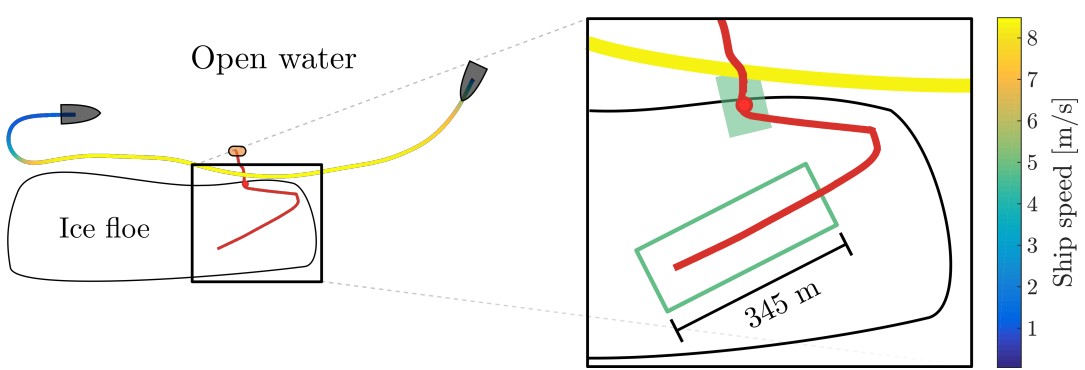

**Figure 2.** Schematic representation of the NBB experiment. The grey shape represents the ship and the orange ellipse the zodiac. The red and blue-to-yellow lines are respectively the UAV and ship GPS tracks. The solid green rectangle is the area of the panoramic picture selected for analysis (Figure 8). The approximate shape of the ice floe is also shown.

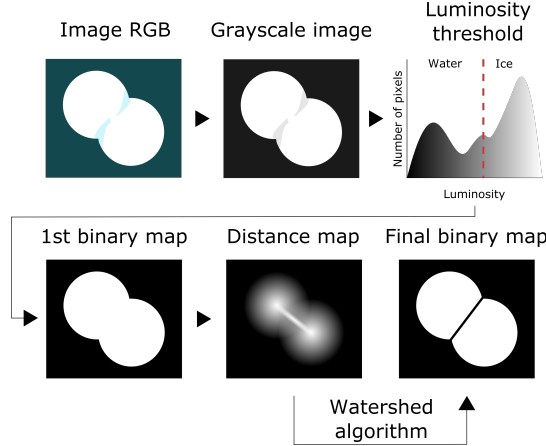

**Figure 3.** Steps of the image processing algorithm used for the boundary identification of floes in the GSL experiment.

distance from the ice floe to launch the UAV. Unfortunately, overwash from the primary wave pushed and flipped the buoys such that their data is unusable. The propagation of flexural waves in the ice floe could not be observed visually due to the flat lighting conditions of an overcast sky. No in-ice wave properties could be measured. The UAV took pictures of the fragmented
ice floe after the passage of the waves. These pictures were used to generate an orthomosaic picture of a portion of the broken floe using the open source software Open Drone Map (ODM), with a resolution of 5 cm px$^{-1}$.



# 3 Image processing

## 3.1 Sea ice segmentation

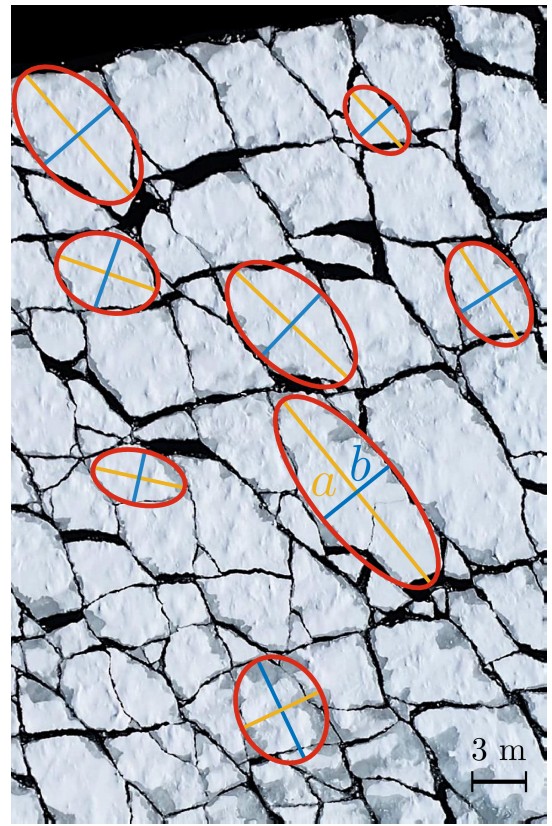

**Figure 4.** Sample of the ellipses fitted on the ice floes by Maltab. Yellow and blue lines indicate the major and minor axes of length $a$ and $b$, respectively.

The detection of sea ice floes in each individual video frames consists in a series of steps that are illustrated in Figure 3.

First the RGB image is converted to grayscale, with values from 0 to 255. From the histogram of the grayscale image, an intensity threshold separating dark water pixels from bright ice pixels is used to create a binary images where 1 indicates ice and 0 indicates water. For many reasons, this binary map does not perfectly separate ice and water, and each ice floe from their neighbours. The presence of slush or very small ice fragments in between floes can merge many floes into a larger one. Conversely, wet ice can generate holes in floes. To circumvent these imperfections, various segmentation algorithms have been

developed and applied in similar contexts, namely the morphology gradient (Zhang et al., 2012), the watershed transform (Meyer, 1994; Zhang et al., 2013) or the gradient vector flow (Zhang and Skjetne, 2014, 2015). We refer the reader to (Zhang and Skjetne, 2018) for a good review of image processing methods that are applicable to sea ice segmentation.





Here, the method that gives the best results is the watershed transform (Meyer, 1994). It uses the Euclidean distance transform of the first binary map to generate a topographic map for each individual object. When two objects are in contact, the two
watersheds connect through a *valley*. If this valley is deeper than some chosen threshold depth, then the two watersheds are segmented across that valley and the original floe is separated in two. This method can sometimes generate new floes, under-segment or over-segment existing floes (Zhang and Skjetne, 2018), but it is possible to circumvent these issues with fine adjustments. To avoid over-segmentation, local minima in the distance transform are removed. To avoid under-segmentation, successive morphological erosion and dilation steps are applied to separate floes that are in contact. Finally, only objects with
an isoperimetric ratio $\Gamma = P^2/4\pi A$ smaller or equal to 3.5 are identified as floes for further analysis.

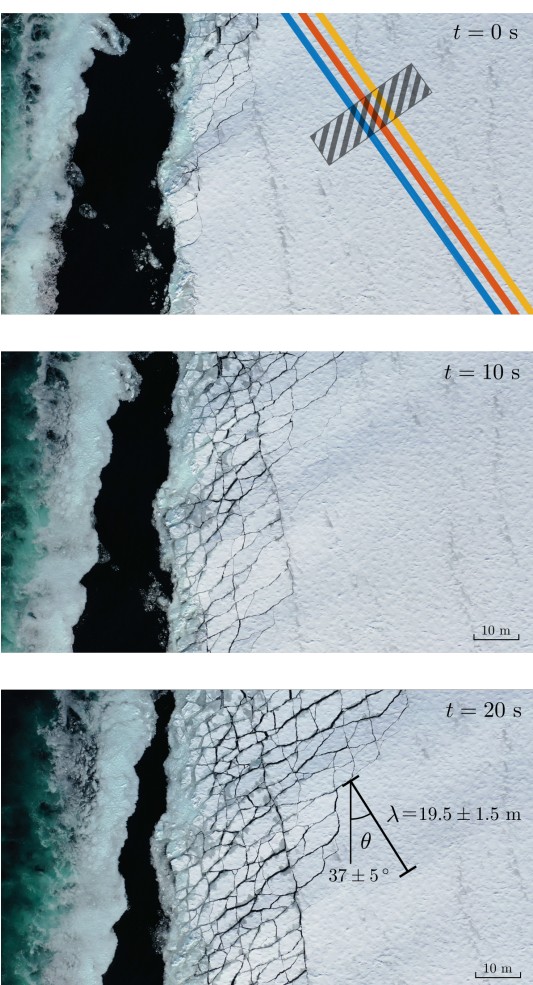

**Figure 5.** Snapshots of the wave-induced break-up experiment at $t = 0$, $t = 10$ s and $t = 20$ s. Flexural waves are visible in the ice prior to break-up with $\lambda = 19.5 \pm 1.5$ m. Colored polygons in the top panel were used to calculate the break-up speed, and the dashed rectangle is the region of interest (ROI) used for the estimation of wave period.





Similarly to Herman et al. (2018), floes having a horizontal dimension that is close to their thickness are removed, since they can't be generated by flexural failure. In the GSL experiment, $h \simeq 30$ cm. Thus, floes having a surface area $A \leq 900$ cm$^2$ are removed. For the NBB experiment, $h \simeq 60$ cm and floes with $A \leq 3600$ cm$^2$ are removed. A thorough visual examination of the final binary image of the GSL experiment shows that it corresponds well to the initial image so that morphological

properties of sea ice floes can be extracted from it. For the NBB experiment, the omnipresence of large melt ponds at the surface of the floes made it impossible to detect efficiently sea ice boundaries using the watershed transform method. Images are thus manually segmented. Using the Matlab Image Processing Toolbox, numerous morphological properties of sea ice floes are extracted. For instance, each floe is fitted to an ellipse of major axis length $a$, minor axis length $b$ (Fig. 4), which is indicative of the distance between wave-induced cracks. Following Squire (1995) and Herman et al. (2018), the minor axis

length is the chosen floe length scale as it represents the characteristic break-up length scale.

## 3.2   Break-up evolution

Using drone footage in the GSL, it was possible to estimate the wavelength and the wave period due to favorable lighting conditions provided by the oblique sunlight. The wavelength $\lambda$ is obtained from the visual estimation of the distance between two consecutive wave crests. The period $T$ is obtained from a Fourier transform of the time variations of the mean brightness

of a region of interest having a size less than half a wavelength and aligned perpendicularly to the wave direction (dashed rectangle in Figure 5). The wave phase speed is then obtained by $c_p = \lambda/T$.

The speed of the break-up front is evaluated using an automated and unsupervised algorithm that identifies the furthest fracture point along each of the colored lines in the direction of wave propagation shown in Figure 5. The algorithm computes the brightness as the mean of the image RGB values and finds the furthest pixel having a brightness value under 90, the

maximum being 255. This threshold is chosen manually after verification with a limited number of frames, and the algorithm is applied every 0.2 s. The break-up speed is then obtained as the slope of the linear regression relating fracture distance to elapsed time.

## 4   Results

### 4.1   Waves and break-up

Table 1 presents a summary of the physical parameters that were measured and estimated in both experiments. In the GSL experiment, the wave period measured from brightness variations in the UAV footage is $T = 4.0 \pm 0.2$ s (Fig. 5). The in-ice wavelength is $\lambda_{\mathrm{ice}} = 19.5 \pm 1.5$ m. To estimate the incident wavelength and the in-ice wavelength in the NBB experiment, for which no direct measurements were made, we revert to the first order linear Kelvin ship wake theory (Thomson, 1887). This theory states that the wavelength generated by ship sailing in a straight line at a speed $U$ is

$$\lambda^* = \frac{2\pi U^2}{g} \cos^2 \vartheta \qquad\qquad (1)$$





where $\vartheta$ is the wave angle with respect to the ship heading and $g$ is the gravitational acceleration. Waves generated from a moving ship can have any angle but waves of maximum amplitude ($a_{\max}$) propagate at an angle of $\vartheta\big|_{a_{\max}} = \sin^{-1}\left(1/\sqrt{3}\right) \simeq$ 35.26° (Soomere, 2007), which is within the uncertainty range of our measurement. Using this angle, the incident wavelength (in-water) for both experiments are thus $\lambda^*_{\mathrm{GSL}} = 22.5$ m and $\lambda^*_{\mathrm{NBB}} = 29.9$ m. Using the deep water dispersion relation, we can

obtain the wave period for both cases $T_{\mathrm{GSL}} = 3.8$ s and $T_{\mathrm{NBB}} = 4.4$ s. These values are going to be useful later in the discussion. Note that for this theory to apply it requires that the sea state is stationary and that the ship speed is constant, which is the case in both experiments.

Figure 5 shows snapshots of wave-induced break-up in the GSL experiment. Flexure-induced cracks are parallel to the wave phase plane, which propagates at an angle of 37°±5 with respect to the ship heading. The three colored step-like curves on

figure 6 show the position of the furthest crack along the corresponding line shown on figure 5 as a function of time. The break-up speed is obtained as the mean slope of the three linear regression, with a 95% confidence interval, and is $1.86 \pm 0.04$ m s$^{-1}$. This value is approximately 2.5 times slower than the measured wave phase speed $c_p = \lambda/T \simeq 5.0 \pm 0.8$ m s$^{-1}$.

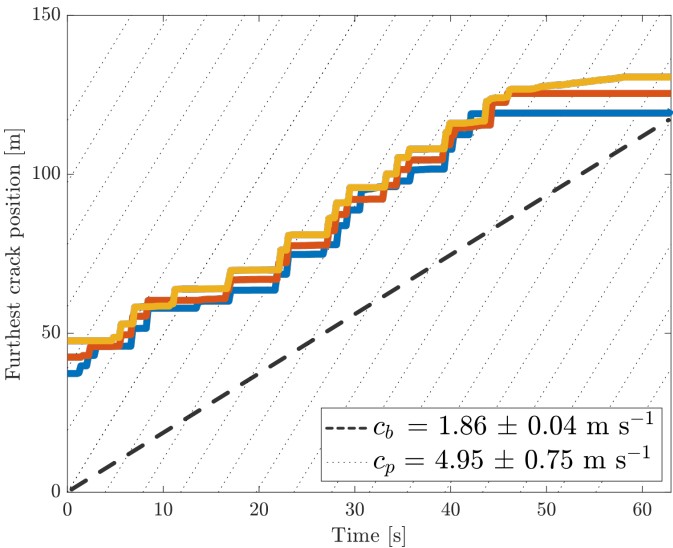

**Figure 6.** Temporal evolution of the furthest crack location relative to the floe edge at $x = 0$ along the three colored polygons shown in Fig. 5. Dotted lines indicate position of wave crests while the slope of the bold dashed line is obtained from the linear regression.

## 4.2  Number-based floe size distribution

In the GSL, the ice fractured up to 60 m from the ice edge leading to a partial break-up of the floe. In the NBB experiment, the

540 m-wide floe completely broke up by the ship-generated waves. Figure 7 shows the breakup that has been captured by the UAV shortly after the passage of the ship (parallel to the horizontal axis of the of the image) in the GSL. Figure 8 shows only a part of the broken-up floe in NBB. The floe was too large to be mapped entirely. In both images, one can distinguish that the





**Table 1.** Wave and ice parameter values measured, estimated or calculated for both experiments, with associated uncertainties. Values in parentheses indicate extreme values whenever they are not equidistant from the mean.

| Parameter | Symbol | GSL | NBB |
|---|---|---|---|
| Ship speed | $U$ | $7.24 \pm 0.08$ m s$^{-1}$ | $8.35 \pm 0.05$ m s$^{-1}$ |
| Wave period | $T$ | $4.0 \pm 0.2$ s | $4.4 \pm 0.2$ s[a] |
| Incident wavelength | $\lambda^{\star}$ | $22.5 \pm 1.5$ m[b] | $29.9 \pm 1.4$ m[b] |
| In-ice wavelength | $\lambda$ | $19.5 \pm 1.5$ m | $24.2 \pm 1.4$ m[c] |
| Incident wave amplitude | $a_0$ | $0.25 \pm 0.15$ | $0.35 \pm 0.15$ |
| Ice floe thickness | $h$ | $0.3 \pm 0.1$ m | $0.5 \pm 0.1$ m |
| Air temperature | $T_{\mathrm{air}}$ | $-7.2$ °C | $+4.9$ °C |
| Ice temperature | $T_{\mathrm{ice}}$ | $-7 \pm 2$ °C | $-3 \pm 1$ °C |
| Ice salinity | $S_{\mathrm{ice}}$ | $5 \pm 1$ | $3 \pm 1$ |
| Brine volume | $v_b$ | $0.04\ (0.02, 0.06)$ | $0.06\ (0.03, 0.10)$ |
| Break-up extent | $L_{\mathrm{MIZ}}$ | $60$ m | $> 560$ m |
| Wave energy attenuation | $\alpha$ | $8.0 \times 10^{-2}$ m$^{-1}$ | $< 1.1 \times 10^{-2}$ m$^{-1}$ |

[a] Estimated using the Kelvin theory Eq. (1) and the deep water dispersion relation $\omega^2 = gk$.

[b] Estimated using Eq. (1) with $\vartheta\big|_{a_{\max}}$ and respective ship speeds $U$.

[c] Estimated using the mass loading dispersion relation Eq. (13).

largest floes have a preferential size, the crack pattern is quite homogeneous and floes have various sizes and shapes with sharp corners. Those are clear signs indicating that they have been broken-up by the bending of surface waves.

To characterize the FSD, let's compute the number-based floe size distribution (NFSD) as it is done in most studies (Rothrock and Thorndike, 1984; Toyota et al., 2006; Herman et al., 2021). As said before, the metric we use for the floe size is the minor axis $b$ in order to relate to a flexural break-up length. To build the distribution, we determine a certain number $M$ of size categories $b_i$, $i = 1 \dots M$. To make sure that the result remains independent of the binning, we follow the normalization proposed by Stern et al. (2018) and set the number of bins $M = \sqrt{N}$, $N$ being the total number of floes. The constant bin

width $\Delta b$ is set equal to the size range divided by the number of bins, i.e. $\Delta b = (b_{\max} - b_{\min})/M$. The number-based normalized probability distribution is then given by

$$P_N(b_i) = \frac{n_i}{N\Delta b}, \qquad \sum_{i=1}^{M} P_N(b_i)\Delta b = 1, \tag{2}$$

Figure 9 shows the NFSDs resulting from both experiments. The GSL NFSD exhibits a modal shape with a mean value of 2.8 m a standard deviation of 1.2 m. The NBB NFSD on the other hand has a bimodal shape with a mean value of 5.9 m and a

standard deviation of 3.4 m. It is interesting to see that the NBB FSD as a shape similar to what Herman et al. (2018) obtained in a laboratory wave-induced ice fracture experiment (more precisely in their test A 2060), which was interpreted as the sum of a power-law and a Gaussian distribution. However, when *looking* at orthomosaics in Figs. 7 and 8, one can see that the area is



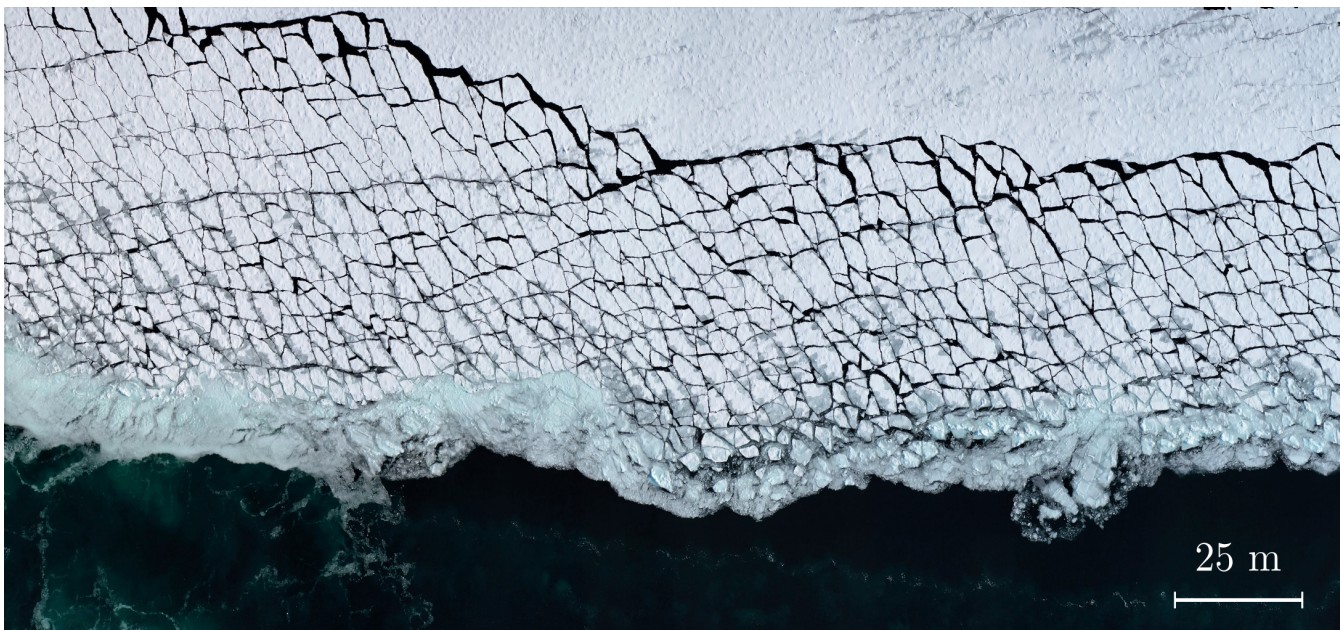

**Figure 7.** Break-up resulting from the GSL experiment. The ship sailed along the horizontal axis of the image.

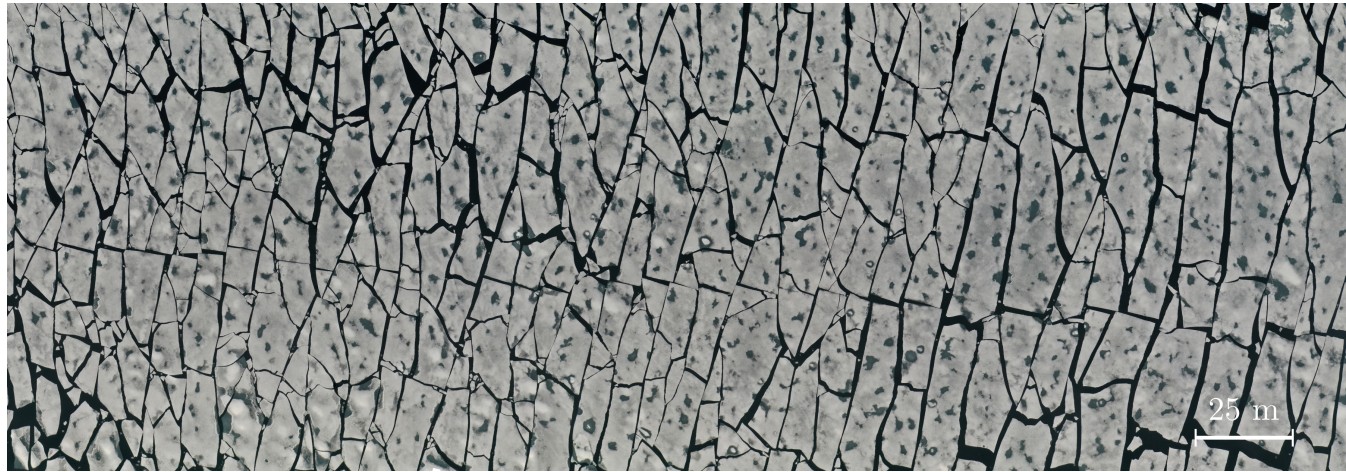

**Figure 8.** Partial view of the break-up resulting from the NBB experiment, corresponding to the green rectangle in Figure 2.

mostly covered by large floes having a size that does not vary a lot. There are indeed, looking closely particularly at the NBB orthomosaic, a large number of very small pieces. This apparent disagreement comes from the fact that the NFSD weighs each

floe equally, whereas a visual assessment puts more weight on larger floes because they cover a larger portion of the image than small floes. The next section presents an alternative way for computing the FSD that is more representative of how floes are organized in space, i.e. floating at the surface of a flat ocean and showing their wider face.





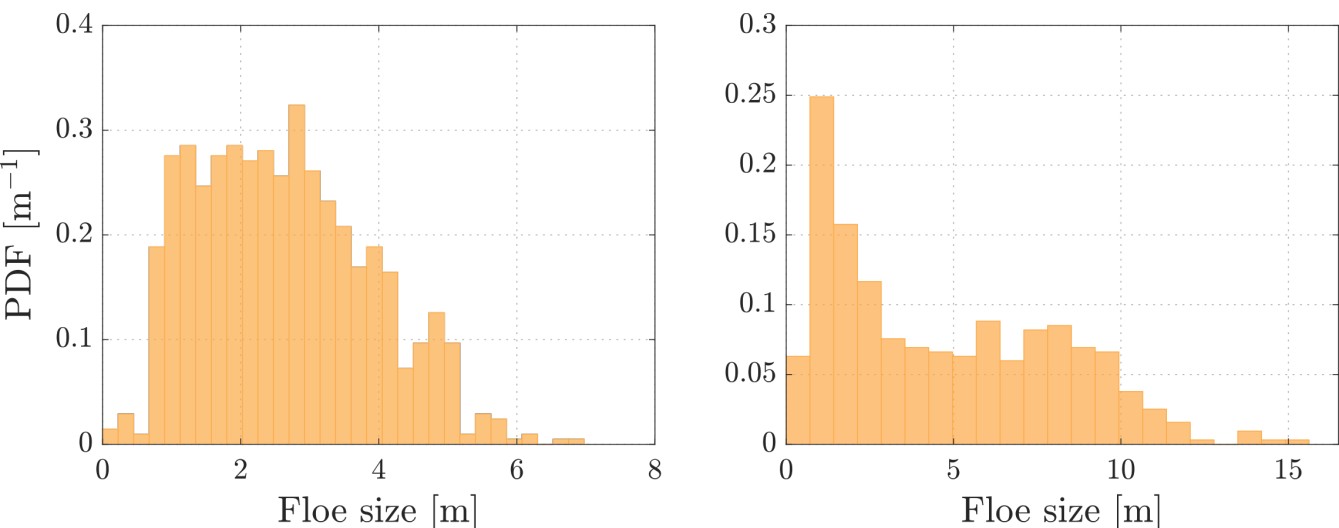

**Figure 9.** Number-based probability density functions of floe size (NFSDs) resulting from the break-up experiments.

## 4.3 Area-based floe size distribution

Let $A$ be the total area covered by ice floes, given by $A = \sum_{k=1}^{N} a_k$, with $a_k$ the area of floe $k = 1, ..., N$. The probability that

a given ice pixel in the image belongs to floe $k$ of area $a_k$ is $a_k/A$. The larger the floe, the higher the probability. Following the
same procedure as for the number-based PDF for the binning, the area-based probability is obtained by

$$P_A(b_i) = \frac{1}{\Delta b} \sum_{j=1}^{n_i} \frac{a_j}{A}, \qquad \sum_i P_A(b_i)\Delta b = 1, \tag{3}$$

where $a_j$ is the area of the $j^{\text{th}}$ floe belonging to the $i^{\text{th}}$ size bin. This representation is not only compatible with a visual
evaluation of an arrangement of flat plates on a surface, it is also compatible with the definition of the ice thickness distribution

(ITD) that is widely used in sea ice models to characterize the state of the ice cover over a given area. The ITD describes the
area of the ice cover that belongs to a given ice thickness category, and the area-based FSD defined by Eq. 3 describes the area
of the ice cover that belongs to a given floe size. The fact that the area-based description can be constrained by the total ice
area is another advantage versus adopting a number-based description which is a priori unbounded.

Figure 10 shows the AFSD for both experiments. By comparing Figures 9b and 10b to Figure 8, it is clear that the area-based

method is more representative in terms of area coverage than the number-based distribution, since it gives more weight to large
floes. Compared to Figure 9a, Figure 10a shows a slight shift of the mean towards higher values. It is worth noting here that
the result is much more robust to segmentation errors than the NFSD since small artefact floes generated during segmentation
are given a very small weight.





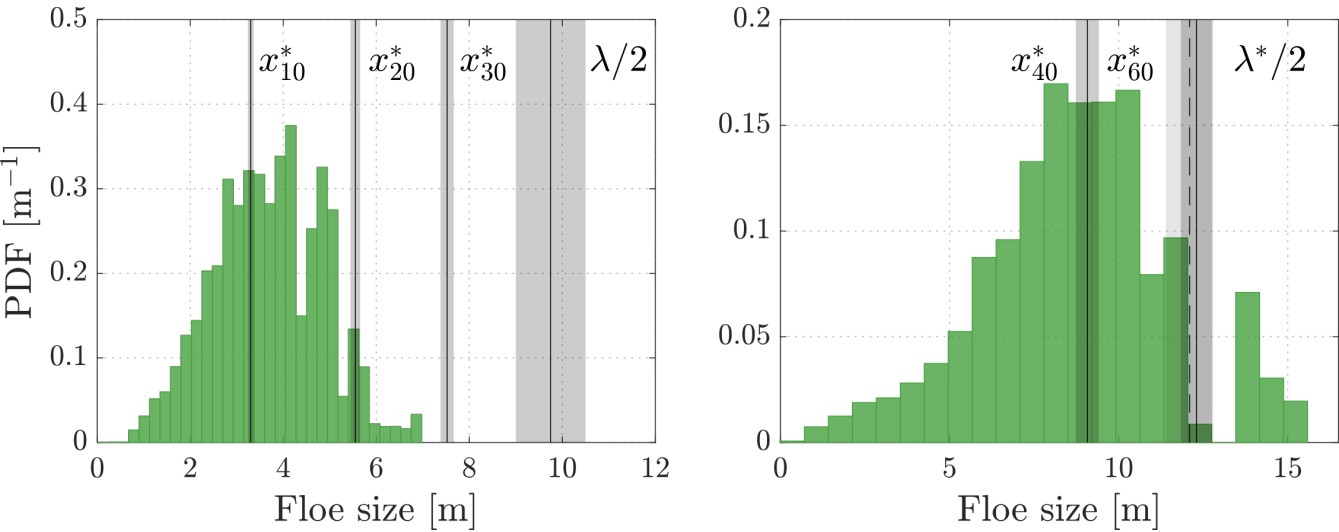

**Figure 10.** Area-based floe size distributions (AFSD) resulting from the break-up experiments computed with Eq. (3). $\lambda$ is the observed wavelength in ice in the GSL, $\lambda^*$ is the wavelength of the ship-generated wave estimated using Eq. (1), and $x^*$ is the flexural rigidity length scale given by Eq. (8) for different thickness values, indicated by indices in cm, with gray bars indicating uncertainties. All floes are considered.

## 5 Discussion

There are very few reports in the literature on wave-induced sea ice break-up events, and even fewer that are observed at adequate time and spatial scales. The two experiments presented here carried out in two contrasting sets of conditions shed light on multiple aspects of wave-ice interactions: 1) the floe size distribution that results from wave-induced break-up, 2) wave propagation in sea ice and 3) wave attenuation, all of which are thoroughly discussed below.

### 5.1 Wave-induced floe size distribution

The AFSDs obtained in this study (Figure 10) follow Gaussian-like distributions rather than a power law, which means that a preferential size is obtained when sea ice is broken-up by waves. Additionally, the mean value $\overline{D}$ of the AFSD increases with thickness: $\overline{D}_{\mathrm{GSL}} = 3.57$ m with $h_{\mathrm{GSL}} \in [10, 30]$ cm, while to $\overline{D}_{\mathrm{NBB}} = 9.00$ m with $h_{\mathrm{NBB}} \in [40, 60]$ cm, suggesting, like a number of previous studies (e.g. Fox and Squire, 1991; Squire, 1995; Herman, 2017), that the dominant floe size obtained from wave-induced sea ice flexural break-up is controlled by sea ice mechanical properties, rather than the wave length scale (e.g.,
Squire, 1995; Fox and Squire, 1991; Herman, 2017). In order to relate the AFSDs obtained here with a relevant physical length scale, let's recall the two main hypotheses there are in the literature.

The first hypothesis is the one used by Dumont et al. (2011) in their wave-ice interaction model. Their parameterization considered that the maximum floe size is the distance between two consecutive location of maximal flexural strain in an infinitely long ice plate that conform to a monochromatic sinusoidal wave described as $\eta(x,t) = a\sin(\omega t - kx)$. The flexural





strain $\varepsilon$ applied at a given time at the ice surface is therefore

$$\varepsilon = \frac{h}{2}\frac{\partial^2 \eta}{\partial x^2} = -\frac{ak^2 h}{2}\sin(kx - \omega t). \tag{4}$$

where $\omega$ if the wave angular frequency, $k$ is the wavenumber, $a$ its amplitude, and $h$ the ice thickness. Taking the first-order derivative of Eq. (4) and setting it to zero gives the location of strain extrema, which are separated by a distance of $\lambda/2$ (Dumont et al., 2011). This approach assumes that the break-up occurs after the monochromatic wave propagated into an unbroken ice plate, at many places within the ice plate simultaneously, and far from the stress-free edge. However, as will be discussed in section 5.2, break-up occurs near the ice edge and progressively at a speed that is close to the wave group speed thus invalidating this hypothesis.

A more physically accurate representation of the situation of interest is provided by (Fox and Squire, 1991). They numerically computed the strain imposed by regular and irregular wave trains of various characteristics to semi-infinite ice plates of various thicknesses using the thin elastic plate theory. They found that the strain increases from zero at the edge to a value that is proportional to the thickness of the ice, to the amplitude of the wave, and to the curvature at the surface of the wave. The length scale that would determine the floe size resulting from the break-up do not depend on the wavelength, but rather on ice thickness and wave amplitude. Unfortunately, there is no simple analytical solution we can use to scale our result. Instead, we use the analytical solution derived by Mellor (1983), based on the work of Hétenyi (1946), that describes the bending of a semi-infinite plate supported by an elastic foundation. A derivation is recalled here for clarity, and the result is used to scale the AFSDs.

Consider a semi-infinite beam extending along the $x$ axis that is submitted to a load $P$ acting downwards at its edge. This generates a vertical deflection of the beam's edge that imposes a bending moment $M$ defined as

$$M = -EI\frac{\partial^2 \eta}{\partial x}, \tag{5}$$

where $E$ and $I$ are respectively the elastic modulus and moment of inertia of the beam (Hétenyi, 1946). Considering a stress-free condition at the edge and a moment that vanishes as $x \to \infty$, the general solution is

$$M = -\frac{P}{\mu}e^{-\mu x}\sin\mu x, \qquad \mu = \left(\frac{k_f}{4EI}\right)^{\frac{1}{4}}, \tag{6}$$

where $k_f$ is the foundation modulus, which can be viewed as a Hooke's constant, and $x$ is the axial direction of the beam (Hétenyi, 1946). Setting the first-order derivative of Eq. (6) to zero, we obtain the following algebraic equation

$$e^{-\mu x}(\cos\mu x - \sin\mu x) = 0, \tag{7}$$

which is satisfied when $x \to \infty$ or when $x = (4n+1)\pi/4\mu$ with $n = 0, 1, 2, ....$ This implies that the location of the maximum bending moment, and therefore of maximal deformation, is

$$x^* = \frac{\pi}{4}\left(\frac{Y h^3}{3\rho_w g(1-\nu^2)}\right)^{1/4}. \tag{8}$$





where we have used the following equivalences for the elastic modulus (Boutin et al., 2018), moment of inertia (Hétenyi, 1946)
and foundation modulus (Boutin et al., 2018)

$$E = Y \,, \quad I = \frac{h^3}{12(1-\nu^2)} \,, \quad k_f = \rho_w g.$$

Here, $Y$ is Young's modulus for sea ice, $h$ its thickness, $\nu = 0.3$ is the Poisson ratio, $\rho_w \simeq 1025 \ \mathrm{kg \ m^{-3}}$ is the sea water density
and $g$ is gravitational acceleration.

Mellor (1983) used this framework for determining a flexure-induced fracture distance in the context of ice rafting, not for
the case of wave-induced breakup. He wrote that "*when the ice is flexed, it will tend to break first at a distance $x^*$ from the free
edge*". This comment has led Toyota et al. (2011) to consider that $x^*$ is "*the minimum ice length at which breakup will occur
due to flexure stress*", Williams et al. (2013a) to interpret $x^*$ as "*[corresponding] to the diameter below which flexural failure
cannot occur*" and Boutin et al. (2018) to assume $x^*$ is the diameter "*below which [...] no flexural failure is possible*". But,
since WIMs use $\lambda/2$ as the maximal floe size based on the fact that sea ice will break at the extrema of deformation, shouldn't
$x^*$ be also considered as a maximum floe length scale since it lies on the same mathematical premises ?

One key aspect to consider in order to understand that $x^*$ represents a maximal size is material fatigue. It has been shown
by Langhorne et al. (1998) that sea ice subjected to waves that generate strains unable to break the ice but above the endurance
limit generates material fatigue in the ice. The latter causes the ice to break at strains lower than its initial flexural rigidity.
Thus, the ice can break before reaching the peak of deformation. Consequently, $x^*$ represents a maximum length scale for floes
resulting from a wave-induced flexural break-up event.

In order to compute $x^*$ for both experiments we need to estimate the effective Young's modulus and the ice thickness has to
be known. Following Timco and Weeks (2010), the Young's modulus can empirically determined by

$$Y = Y_0(1 - 3.51 v_b) \tag{9}$$

where $Y_0 \simeq 10 \ \mathrm{GPa}$ and $v_b$ is the brine volume. As argued by Williams et al. (2013a), an effective value of $Y^* = Y - 0.5 \ \mathrm{GPa}$
must be used when a cyclic loading of a period less than 10 s is considered. We thus used $Y^*$ for calculating the flexural rigidity
length scale $x^*$ in what follows since waves in our experiments had a period lower than 10 seconds. The brine volume depends
on ice salinity, but even more strongly on ice temperature. The warmer the ice the larger the brine volume and the porosity.
Here we use the relationship of Cox and Weeks (1983) to estimate the brine volume, which is

$$v_b = \frac{\rho_{\mathrm{ice}} S}{F} \quad \text{with} \tag{10}$$

$$F = \begin{cases} -4.732 - 22.45 \, T_{\mathrm{ice}} - 0.6397 \, T_{\mathrm{ice}}^2 - 0.01074 \, T_{\mathrm{ice}}^3 & \text{for} - 2°\mathrm{C} \geq T_{\mathrm{ice}} \geq -22.9°\mathrm{C} \\ 9899 + 1309 \, T_{\mathrm{ice}} + 55.27 \, T_{\mathrm{ice}}^2 + 0.716 \, T_{\mathrm{ice}}^3 & \text{for} - 22.9°\mathrm{C} > T_{\mathrm{ice}} > -30°\mathrm{C} \end{cases} \tag{11}$$

Unfortunately, the salinity of the ice was not measured during the experiments. To estimate it, we recall the empirical study of
Cox and Weeks (1974) who show that thin and cold young ice has a typical salinity around 5 ppt, while warm sea ice at the
end of the melt season can have as low as 2 ppt. We thus set $S_{\mathrm{GSL}} = 5 \pm 1$ and $S_{\mathrm{NBB}} = 3 \pm 1$ to account for the two different



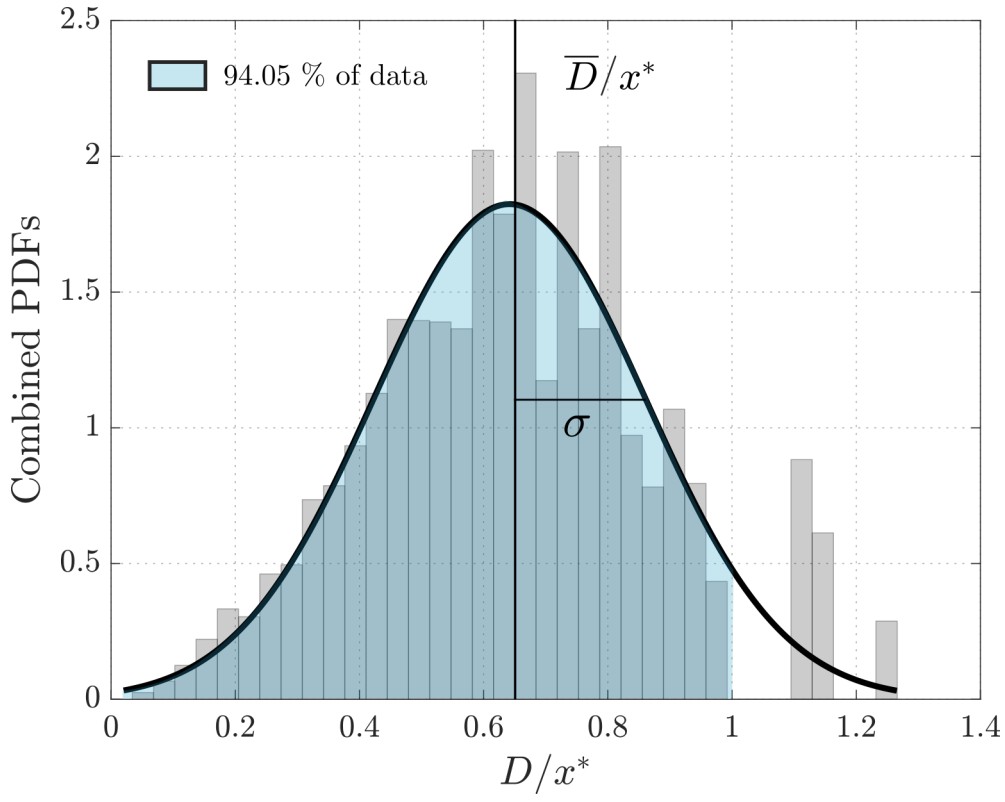

**Figure 11.** Area-based floe size probability density functions (PDF) as a function of the normalized floe size showing results from both experiments (data of Figures 10a and b). The black line is a least square-fitted Gaussian distribution.

situations as well as the uncertainty associated to these estimates. The ice temperature is set close to the freezing point of

sea water for NBB, and close to the air temperature for GSL (see Table 1). This gives $v_b = 0.04$ $(0.02, 0.06)$ for GSL and $v_b = 0.06$ $(0.03, 0.10)$, with extrema values shown in parentheses. Using Eq. (9) with these parameter values, the maximum and minimum values for the effective Young's modulus values in the GSL experiment are $[Y^*_{\min}, Y^*_{\max}] = [7.93, 9.23]$ GPa and are $[Y^*_{\min}, Y^*_{\max}] = [6.56, 8.85]$ GPa in the NBB. Knowing that sea ice in the GSL experiment is between 10 and 30 cm, values of $x^*$ computed with $h = 10,\ 20$ and $30$ cm are shown (respectively $x^*_{10}$, $x^*_{20}$ and $x^*_{30}$) in the left panel of Figure 10. The extent

of the shaded areas is determined by the minimum and maximum values of $x^*$ given the range of Young's modulus for a fixed thickness and the solid line is the mean value. The same thing is shown for NBB in the right panel of Figure 10 with thickness values of 40 cm $(x^*_{40})$ and 60 cm $(x^*_{60})$.

     The AFSD of GSL (Fig. 10a) and NBB (Fig. 10b) experiments are compared to both $\lambda/2$ and to $x^*$ in order to see if these quantities could represent the maximum floe size. Figure 10 shows that $x^*$ is more sensitive to thickness than to Young's

modulus. But, based on the fact that $x^*$ and $\lambda/2$ should both represent the maximal floe length due to flexure and since it is the only mechanism which can have caused the largest floes in our data – thermal cracking and fracture by isostatic adjustment





have not occurred during the experiments – there should be very few floes larger than $x^*$ or $\lambda/2$. This statement allows to conclude that the thickness of the ice over which the images were recorded in the GSL experiment might have been close to 30 cm while for the NBB, it should have been over 60 cm. Note that in the NBB experiment, ice thickness was not measured

directly in the area where images were recorded to create Figure 8. Before the fragmentation, only the edge of the initial floe could be measured with a value of 3 to 5 cm. After the break up however, we measured thickness values between 40 and 60 cm for floes further in. There was therefore a horizontal variation of thickness within the ice plate used for the NBB experiment. We thus think it is highly probable that the floes in Figure 8 had a thickness greater than 60 cm and hence that $x^*$ might be even closer to the observed maximal floe size in this experiment.

In order to propose a functional form to the FSD contribution from wave-induced breakup, we combined both AFSDs and fitted a Gaussian function to the data. The GSL and NBB AFSDs were respectively normalized by their *modelled* maximal size, namely $x_{30}^*$ and $x_{60}^*$. The width of the bins was calculated the same way than described before but by considering the minimal and maximal normalized sizes from both distributions. The number of bins was calculated with the total number of floes in the two experiments. The resulting AFSD is shown in Fig. 11. The black curve is a parametric fitting of a Gaussian

function, namely

$$\phi(d) = \frac{1}{\sigma\sqrt{2\pi}}e^{-(d-\overline{d})^2/2\sigma^2} \tag{12}$$

where $d = D/x^*$ is the adimensional floe size, $\overline{d}$ is the mean adimensional floe size and $\sigma$ is the standard deviation with $\overline{d} = 0.6418$ and $\sigma = 0.2187$ at a 95% confidence interval. The fact that $x^*$ covers 94.05 % of the data in both experiment suggests that the FSD resulting from wave-induced breakup can be modeled as a Gaussian distribution. We propose that it

should be lower bounded by the ice thickness $h$, since no floe can be smaller than its thickness, and upper bounded by $x^*$ as a maximum size. These results highlights the fact that the power law is not representative of the floe size distribution produced by wave-induced break-up events, which was so far assumed in some models (e.g. Dumont et al., 2011; Williams et al., 2013a; Zhang et al., 2015).

## 5.2 Break-up evolution and wave propagation

In the two experiments, ice broke-up from the edge inward with the furthest crack at a given time oriented parallel to the wave phase plane. The speed at which the break-up progressed could only be measured in the GSL experiment using the UAV footage. The speed at which it progressed $c_b$ was slower than the phase speed of the wave $c_p$ estimated also using the UAV imagery. Considering the uncertainties, the ratio between the two is $r = c_b/c_p = 0.41 \pm 0.07$, which is close to $1/2$, the ratio of the group speed to the phase speed for waves traveling in deep water. However, here waves travel in sea ice, first in a long

(compared to wavelength) consolidated ice sheet of thickness $h$ and second, into a set of smaller fragments of that same sheet after it broke. A modification of this ratio is thus expected as waves travel in ice with a different dispersion relation. Figure 12 shows the group-to-phase speed ratio for two dispersion relations, the first one describing flexural waves propagating in a thin elastic plate and the second one describing waves affected by the inertia of a floating material, usually called *mass loading*, and





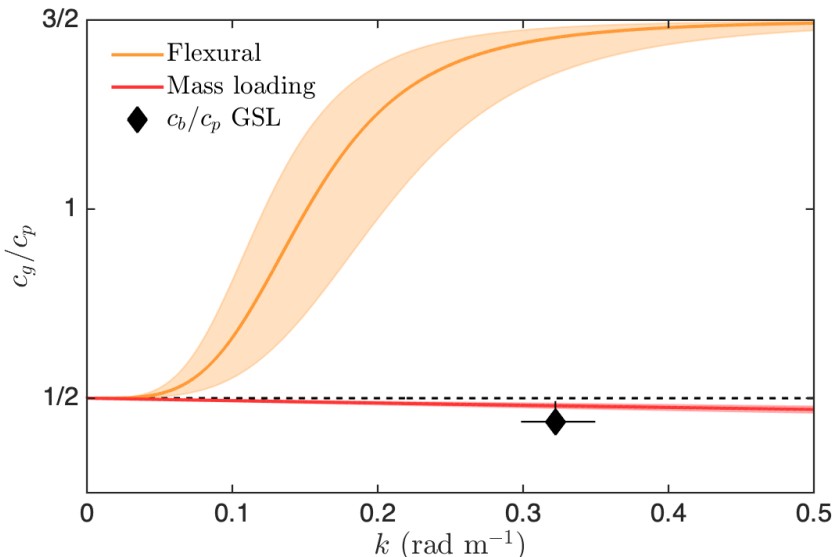

**Figure 12.** Ratio between the group speed and the phase speed of waves traveling into sea ice of thickness $h = 0.3 \pm 0.1$ m and $v_b = 0.05$ as a flexural wave or influenced by mass loading. The black diamond indicates the ratio between the break-up speed $c_b$ and the measured phase speed $c_p$, with associated uncertainties described in Table 1.

given by

$$\omega^2 = gk\left(1 + \frac{\rho_i h k}{\rho_w}\right).$$  (13)

It shows quite clearly that at this frequency, waves that are responsible for the break-up and that are visible from the UAV propagate following the mass loading dispersion relation. The break-up speed $c_b \simeq 1.86 \pm 0.04$ m s$^{-1}$ is only slightly slower than the group speed, i.e. the speed at which wave energy propagates $c_g = \frac{\partial \omega}{\partial k} \simeq 2.13 \pm 0.18$ m s$^{-1}$, but still within the uncertainty interval. In comparison, the group speed of the corresponding flexural wave is on the order of 60 m s$^{-1}$.

## 5.3 Break-up extent and wave energy attenuation

Since waves attenuate as they propagate in sea ice, break-up eventually stops. In the GSL, the break-up extent reached 60 m from the original floe edge, while in the NBB experiment it fragmented the entire floe that was approximately 540 m wide. In the latter, the break-up extent could have been larger if the floe was wider and the value should be regarded as a lower bound. Using the available information and some assumptions detailed below, it is possible one one hand to estimate the flexural strain imposed by waves at the ice edge. On the other hand it is also possible to compare that value to the critical flexural strain, below





which fracture does not happen, to estimate the wave attenuation in both experiments and to relate the attenuation coefficients to the ice conditions. This is what we do below.

The maximum flexural strain induced by a monochromatic wave is $\varepsilon_{\max} = \frac{1}{2}ahk^2$ (see Eq. 4). While the frequency is conserved, both the wavenumber and the amplitude are modified when passing from one medium to the other, according to the

dispersion relation. Based on the previous discussion, we use the mass loading dispersion relation when the wave propagates in unfractured ice. It is unclear whether the same relation applies to fragmented ice or if it rather follows the deep water relation, but as we will argue later it is of negligible importance to our conclusions. The mass loading dispersion relation is such that it reduces the wavelength and, to conserve energy, increases the wave amplitude.

The wave amplitude was not directly measured. It was only estimated visually during the NBB experiment from the zodiac

before it hit the ice floe. The largest wave was slightly less than one meter high. For the following calculation, we thus use an amplitude (half the height) of $a_{\mathrm{NBB}} = 0.35 \pm 0.15$ m to which we associate a large uncertainty. Since the ship speed was higher during the NBB experiment than during the GSL experiment, we chose a smaller value but with a comparable uncertainty $a_{\mathrm{GSL}} = 0.25 \pm 0.15$ m.

When the flexural strain reaches a critical value $\varepsilon_c$, the ice breaks-up. This value depends on the flexural strength of the ice

$\sigma_c$ and on the effective Young's elastic modulus $Y^\star$ through (Williams et al., 2013a)

$$\varepsilon_c = \frac{\sigma_c(1-\nu^2)}{Y^*}. \tag{14}$$

To estimate the flexural strength, we use the empirical relationship of Timco and Weeks (2010)

$$\sigma_c = \sigma_0 e^{-5.88v_b} \tag{15}$$

where $\sigma_0 = 1.76$ MPa. Figure 13 shows the flexural strain induced in the floe by the in-ice wave for both experiments. Unsur-

prisingly, it is significantly larger than the critical strain indicated by the black line. The uncertainties both on brine volume and on the strain values are determined quite conservatively from measurement errors and estimates of unmeasured values. The question we can now try to answer is: how much the wave attenuates so that the strain becomes lower than the critical value? Assuming exponential decay of wave energy with distance, the damping rate $\alpha$ is calculated using

$$\alpha = \frac{1}{x_b} \ln\left[\left(\frac{a_0}{a_b}\right)^2\right] \tag{16}$$

where $x_b$ is the distance traveled by the wave from the original floe edge up to the end of the broken-up region. The amplitude $a_b$ that will cause a critical strain is obtained , given by

$$a_b = \frac{2\sigma_c}{hk^2Y^*}. \tag{17}$$

Table 1 reports estimated values for wave energy attenuation for both experiments: $\alpha_{\mathrm{GSL}} \simeq 8.3\ (6.5,\ 9.3) \times 10^{-2}$ m$^{-1}$ and $\alpha_{\mathrm{NBB}} \simeq 1.07\ (0.94,\ 1.15) \times 10^{-2}$ m$^{-1}$. Even though the ice is thicker in the NBB experiment, the attenuation is almost one

order of magnitude weaker than the attenuation in the GSL. Morever, since the break-up extent for NBB could have been larger,

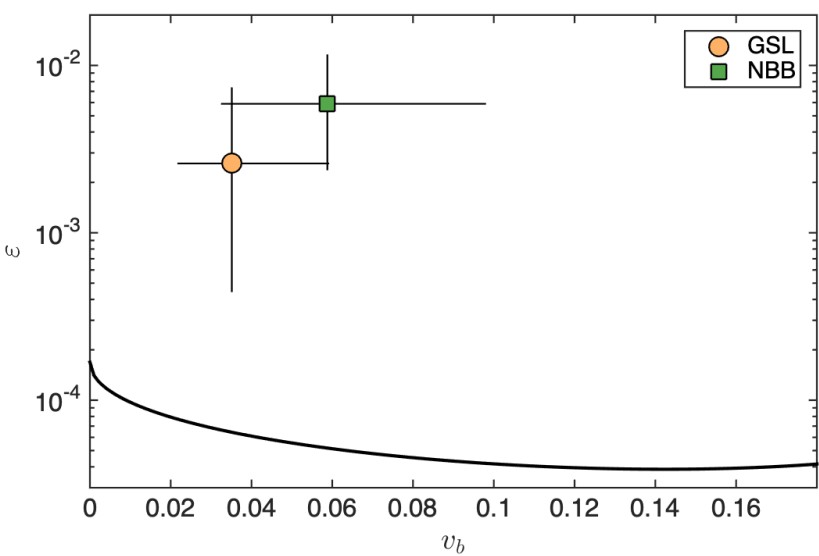

**Figure 13.** Flexural strain induced near the original floe edge by the in-ice wave for both experiments computed by Eq. (4) using values and uncertainties of Table 1. The black line shows the critical flexural strain as a function of brine volume.

this estimation represents an upper bound and the actual attenuation is probably larger. Many processes can cause wave energy to attenuate. Boutin et al. (2018) provides an in-depth sensitivity analysis of many processes, some of which have a dependence on ice thickness, floe size and ice temperature that may explain this result. For example, dissipation through anelastic strain in the ice is highly dependent on ice thickness. For $T \sim 4$ s waves, the attenuation is significantly larger for thinner ice, although the absolute values are one order of magnitude lower than observed. Anelastic and inelastic dissipation strongly depends on floe size: when floe are small compared to the wavelength, flexural strain become negligible. Concluding about the causes of wave attenuation, identifying the processes involved and quantifying their contributions is beyond the scope of this paper. However, our results certainly highlight the potential of using ship-generated waves in conjunction with remote and in situ measurements of waves and sea ice to deepen our understanding of wave-ice interactions.

## 6 Conclusions

Results obtained from the analysis of two wave-induced sea ice break-up experiments, captured by an unmanned aerial vehicle and carried out in two contrasting sets of conditions, provide direct and detailed measurements that shed light on many aspects related to wave-ice interactions. We believe that this approach, combined with in situ measurements of waves and ice properties,



has a great potential to confirm (or invalidate) hypotheses formulated by numerous theoretical or empirical studies based on
indirect observations.

The main conclusion of our study is that the size of floes resulting from wave-induced break-up, obtained with the area probability instead of the number probability, is characterized by a modal distribution, i.e. that floes have a preferential size. This resonates with many other anecdotal observations reported in the literature (e.g. Fox and Squire, 1991; Squire, 1995; Herman, 2017). The other conclusion is that this distribution is constrained by an upper bound that is proportional to the
flexural rigidity length scale. This result contrasts with the interpretation put forward by Toyota et al. (2006) in which $x^*$ was rather acting as a lower bound, or a minimum size for ice fragments. There is no indication that a lower bound exists according to our results. However, setting the smallest size equal to floe thickness seems to be a reasonable choice. The modal shape of the FSDs informs us that sea ice breaks up systematically at strains lower than the extrema such that material fatigue is of important when considering breakup (Langhorne et al., 1998). The spread of the distribution also highlights the fundamental variability
in this process, which is unsurprising considering the ubiquitous heterogeneity of sea ice inherited from complex life cycles. When based on the surface of floes rather than their number, probability density functions are an appropriate representation of the FSD over which a functional form is proposed (Eq. 12. This parameterization uses the area-based FSD for two main reasons: 1) it is more coherent with the spatial organization of ice floes on a surface than the number-based FSD is, and 2) it is coherent with the definition of the ice thickness distribution that is commonly used in sea ice numerical models.

The aerial imagery of the break-up event also allowed the characterization of wave propagation, break-up evolution and extent. First, the break-up advances progressively from the floe edge inward at a speed that is close to the group speed of the wave.

Second, waves propagating in the unbroken ice floe that were captured by the UAV follow the mass loading dispersion relation, such that the wavelength and the phase speed is smaller than in deep water. Flexural waves might propagate, but with
an amplitude such that it could not be detected and that did not cause significant deformation of the ice cover.

Finally, waves were attenuated much faster in the thinner ice floe. Wave energy dissipation and dispersion through inelastic and anelastic deformation of unbroken ice could explain this result. However, the lack of in situ data about sea ice properties and wave characteristics along over the course of their propagation does not allow us to identify the processes at play or to partition the contributions in broken and unbroken ice.

Overall, these results show that using a ship to generate waves and operate in a controlled yet natural environment is a promising way to study the wave energy budget during a wave-induced break-up event and advance our understanding of wave-ice interactions and marginal ice zone dynamics. Collecting key in situ data to complement the UAV could significantly improve the potential. It could be done by using ice-going platforms such as an ice canoe to deploy wave buoys and measure ice properties (thickness, temperature, salinity, snow cover) as what was done by Sutherland and Dumont (2018).

*Code and data availability.* The code and data required to produce the results are available on demand at the following git repository: https://gitlasso.uqar.ca/dumael02/breakup.



*Video supplement.* The video of the GSL experiment is available on ResearchGate at the following https:/doi.org/10.13140/RG.2.2.32873.62564 under a CC BY-NC-ND 4.0 license.

*Author contributions.* EDL planned and conducted both experiments, developed the image processing tools, performed the data analysis, wrote the first draft of the paper and participated to the discussion and writing. DD provided the original idea for the experiment, provided guidance in data analysis, and participated to the discussion and writing.

*Competing interests.* The authors declare that they have no conflict of interest.

*Acknowledgements.* The authors would like to thank the Canadian Coast Guard crew for their collaboration in carrying both experiments on opportunistically. This work was financially supported by the NSERC Discovery Grant to DD *Physics of Seasonal Sea Ice* (RGPIN-2019-06563, RGPAS-2019-00068), the Odyssée Saint-Laurent program of the Réseau Québec maritime and by the Canadian Foundation for Innovation and Amundsen Science.



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
