# Peer review of "Aerial observations of sea ice break-up by ship waves"

_The Cryosphere, 2021_

## Referee Comment (RC1)

**Aerial observations of sea ice break-up by ship waves**

**by Elie Dumas-Lefebvre and Dany Dumont**

General comments

This is an enjoyable little paper which reports some unique results on a topic of meaningful contemporary interest that has been reinvigorated by the effects of global climate change on the polar and subpolar seas. The work is reported clearly and is mostly well-written. Although the observations and subsequent analysis are limited in extent, they are interesting and thought-provoking, as they intersect earlier theoretical conjectures about the way sea ice fractures under the action of ocean waves – namely in regard crack separation and the FSD configuration that is created as a consequence.

Nonetheless, sometimes assertions are made in the paper that are not adequately justified by evidence. The authors should be careful of this and ensure that what they are saying is correct and that earlier papers are cited where applicable as conclusions have been reached in some cases without the benefit of results reported in a previously published paper that remains uncited. This notwithstanding, and appreciating that keeping track of every publication in a research field is impossible, in the reviewer's opinion a revision of the work to accommodate the specific comments and technical corrections below will improve the paper sufficiently for it to be published in *Cryosphere*. I am aware that my review will seem pernickety and possibly irksome at first sight but my intention has been to finesse the paper because I believe its results should ultimately be disseminated through publication.

In sum, I feel that the authors are being too assertive in regard to the outcomes of their study. The methods they have used are novel, exciting and have tremendous potential. I congratulate them for collecting and compiling a manuscript to publish these observations. I am inclined to disagree with some of their unequivocal affirmations which, to me, are not always based on robust evidence and perhaps to some extent show a lack of deep understanding of this multifaceted research topic.

I recommend that the manuscript should be published but only after the authors have acted on the comments below.

Specific comments

1. Line 3–4. What does 'When represented as probability density functions weighted by the surface of ice floes' mean? That is, how can something be weighted by a surface?
2. Line 9 the 'mass loading dispersion relation' won't be known to many readers. I am assuming a citation is not allowed in the abstract, so a definition or explanation should be provided here.
3. Line 9–10. 'Moreover, our experiments show that thicker ice can attenuate wave less than thinner ice'. This is counterintuitive so, if the authors still believe this to be the case after reading my comments, I would suggest adding a short statement explaining why, e.g. Moreover, our experiments show that thicker ice can attenuate waves less than thinner ice, because …'. Notice also that 'wave' has been altered to 'waves'.

4. Line 18. What do the authors mean by 'larger' waves? Waves of greater amplitude or waves of longer wavelength, or both?

5. Line 20. Why do waves 'change the mechanical properties of the ice'? Are the authors hinting at fatigue or are they using the word 'ice' loosely, meaning the ice-cover as a whole?

6. Line 34. Don't the usual definitions of anelastic and inelastic designate that anelasticity is a particular type of inelasticity; if so why do the authors list them separately suggesting that they represent independent material behaviors.

7. Line 36. I find it hard to believe that 'there are no observations that directly relate the FSD to a given process in the natural environment', but accept what the authors intended in the sentences that follow in the text.

8. Line 67. 'This means that wave-induced break-up leads to a bell-shaped FSD, a result that indicates that the morphology of floes resulting from breakup might not be well represented by a power law.' Didn't **Montiel, F. and Squire, V. A. Modelling wave-induced sea ice break-up in the marginal ice zone. Proceedings of the Royal Society of London, Series A 473(2206): 1–25 (2017)** deduce a similar result? This paper should be cited incidentally, as it concerns breakup by waves.

9. Line 70. I personally think the paper **Fox, C. and Squire, V.A. On the oblique reflexion and transmission of ocean waves from shore fast sea ice. Philosophical Transactions of the Royal Society A 347(1682): 185–218 (1994)** is a better one to cite than Fox and Squire (1991), as it is more general and includes everything reported in the 1991 paper, but I accept that the authors are quoting a specific statement.

10. Line 87. The author state 'quantitative analysis is required to fully test these hypotheses' but isn't it obvious that, all things being equal and ignoring any small amplification arising due to the free edge boundary condition, long-crested propagating waves stress all parts of an ice plate similarly, so the specific wavelengths involved will be less important unless standing waves are created.

11. Lines 100–108 are the nub of the paper yet I find the paragraph poorly expressed and somewhat *blah*. As the authors rightly say, it is incredibly difficult to get data on naturally-occurring wave-induced breakup of ice floes because it demands *the stars to be aligned* even when instrumentation is available. What this particular study does is to create the waves and then measure the outcome. The only thing missing is a measurement of the induced curvature or stress. Wouldn't that be nice! At minimum break the paragraph at line 104, i.e. start a new paragraph at 'In this study, we use …' so the reader can immediately see the goals and any limitations of the paper.

12. Line 110–111. What does the word localized mean in 'First, a large level ice floe having a side exposed to open water is localized'?

13. Line 116. In '… hands a better management of weather conditions …', I do not know what 'hands' means in this context. Is this a technical word? Or is it bad English and means 'provides'.

14. On Line 127 the authors state that the sea ice in the Gulf of St. Lawrence was grey and grey-white between 10 and 30 cm-thick'. This is an important observation that the reviewer will refer to later. No sensor was deployed on the ice itself.

15. Likewise lines 140–142 indicate that sea ice in Northern Baffin Bay during the experiment there was heavily rotted first year ice between about 40 and 60 cm. Two SKIb wave buoys deployed on the ice were capsized, unfortunately, so provided no data.

16. Line 205. I note for future reference that the waves are relatively short period, as expected for a Kelvin wake, namely around 4 s.
17. Figures 5–8 are awesome!
18. Line 255–258. I agree with the opinions expressed in this paragraph but the authors must also remind the reader here that the waves being discussed here are of modest period, i.e. around 4 s. Indeed, in regard to natural ocean waves they would be categorized as 'chop'. Furthermore, the sea ice in the experiment is somewhat distinctive, i.e. a subcategory of what is encountered in the polar oceans. It would be disingenuous to suggest that the data being presented are universally applicable to all wave-ice interaction scenarios. This is not intended to lessen the importance of the results being presented, which I regard highly, but rather to ensure than the facts are presented accurately.
19. Eq. (4) assumes the ice floe is an Euler-Bernoulli beam, i.e. the sea ice is thin and homogeneous through its thickness. This should be stated, as sea ice in nature has a temperature and brine volume gradient that renders the latter assumption an approximation.
20. While the statement in lines 278-283 is correct, I do wonder whether the small local peak that occurs near the ice edge is sufficient to explain why the ice progressively fractures from its margin to its interior, given the degree of approximation inherent in the hydroelastic model.
21. Line 283. I am not sure what 'Unfortunately, there is no simple analytical solution we can use to scale our result' is saying. There are a wealth of hydroelastic studies of wave-ice interaction dating back to papers published done in the 1950s and the first Weiner-Hopf analysis of Evans and Davies in 1968 which was generalized in **Williams, T.D. and Squire, V.A. Scattering of flexural–gravity waves at the boundaries between three floating sheets with applications. Journal of Fluid Mechanics 569: 113–140 (2006)**. It is also worth noting that by ignoring the mass of the plate, a fully algebraic result was obtained in **Tkacheva, L.A. Scattering of surface waves by the edge of a floating elastic plate.** Journal of Applied Mechanics and Technical Physics **42(4): 638–646 (2001)**, which could presumably be used. It is disappointing to me that a more *authoritative* model has not been used when so much theory is available; the fragility of the wave-ice interaction topic generally is not situated in mathematical theory but in the paucity of data to validate theory. On the other hand, I am not suggesting that the authors can ameliorate this problem at this stage by reworking their analysis, I am simply conjecturing that a different approach could have changed the manuscript from where I began this review, i.e. 'This is an enjoyable little paper', to a something more substantive.
22. Line 284. I will point out the obvious. The Hétenyi (1946) model has no fluid dynamics, which I would perceive as fundamental to interpreting an ocean wave phenomenon.
23. In Eq. (5), the authors need to say that it is an Euler-Bernoulli beam that they are using as a model. Strictly, they would actually be better to use a Kirchhoff–Love plate, i.e. the plate equivalent of an Euler-Bernoulli beam, which would introduce a Poisson ratio effect. However, both suffer from the obvious approximation that the sea ice is assumed to be homogeneous through its thickness. (I note that most theory assumes a similar paradigm on the basis that the waves are long compared to the thickness, so I am not criticizing the authors for assuming homogeneity, I am simply advising them to tell the reader what they are assuming.)

24. Line 288–291. This is confusing. In consecutive sentences the authors refer to bending moment, moment of inertia and moment. The third occurrence is the bending moment. Why not just say $M$ vanishes for large $x$. And to be pedantic, isn't $I$ the second moment of area rather than the moment of inertia, i.e. no mass?

25. Line 294. 'First-order derivative of Eq. (6)' with respect to $x$.

26. Ah. Eq. (8) suddenly introduces $v$, which we are told later is Poisson's ration. This means that the authors are actually considering a Kirchhoff-Love plate after all, so my comment 23 is superseded. However, I think I could argue that this section needs a good tidy up to be consistent.

27. Line 304. The authors write that 'Mellor (1983) used this framework for determining a flexure-induced fracture distance in the context of ice rafting, not for the case of wave-induced breakup.' This is important as Mellor is using his analysis for a quasi-static problem, while the authors are using it for a dynamic problem. This is why I dislike what they have done.

28. Line 304–310. Rather a dubious argument for all the reasons I have just articulated, namely that the theory being used is inappropriate for the dynamical data set being modeled.

29. Line 313. 'The latter causes the ice to break at strains lower than its initial flexural rigidity.' How can this be? Strain is dimensionless. Flexural rigidity has units of Pa m$^3$. This makes the sentence nonsense.

30. I am rather worried by the arguments used starting at line 316 and 338. I simply don't believe the value of the effective modulus is nearly as high as the authors have estimated. Assuming the arithmetic is correct—and I haven't checked, the argument about the unmeasured salinity is flawed to my mind. The sea ice in question is relatively thin and warm, and the NBB floe is 'heavily rotten' according to the authors, so I would expect a much lower effective modulus. I believe the cited argument about the salinity being less later in the season relates to desalination mechanisms that are not present in warm, highly saline, sea ice of modest thickness (especially acknowledging the uncertainty around NBB thickness, apropos lines 344–349). as sea water at 30–35 ppt flushes a good proportion of the lower parts of the ice matrix. Sea ice is a mushy layer as reported by **Feltham, D. L., Untersteiner, N., Wettlaufer, J. S., and Worster, M. G. Sea ice is a mushy layer, *Geophysical Research Letters* 33: L14501 (2006).** My advice is to think through this section *very* carefully, as the value of effective modulus calculated is way too high in my opinion and, unfortunately, this has a bearing on the subsequent analyses and conclusions. Incidentally, while Cox and Weeks (1983) is undoubtedly the most comprehensive publication on brine volume, an easier analysis was completed earlier by **Frankenstein, G. and Garner, R. Equations for determining the brine volume of sea ice from −0.5° to −22.9°C. Journal of Glaciology, 6(48): 943-944 (1967)**.

31. Line 371–379. The focus of my concern signalled in the previous item-30 targets the issue of whether the sea ice is behaving as a mass loading medium or as a flexible plate, as the latter depends strongly on the value of the flexural rigidity $Y^*h^3/12(1-v^2)$, primarily expressed via thickness $h$ but also the effective modulus $Y^*$. If $h$ or $Y^*$ are over estimated then the dispersion relation will be incorrect. The somewhat uninspiring publication **Squire, V.A. A comparison of the mass-loading and elastic plate models of an ice field. Cold Regions Science and Technology 21:219–229 (1993)** points out why this is so, namely that the mass-loading dispersion relation is just the flexural plate one

with zero flexural rigidity. The authors should take a look at Fig. 7 of that paper, which shows how wavelength is affected by a change of flexural rigidity at a fixed thickness, i.e. a change of $Y^*$. Getting $Y^*$ or $h$ right is crucial. My question is how does it affect the very strong argument for the mass-loading model made in lines 376–379, where the authors state categorically 'It shows quite clearly that at this frequency, waves that are responsible for the break-up and that are visible from the UAV propagate following the mass loading dispersion relation.'

32. Line 381–382. I note here and on line 141 that the NBB floe was 540 m wide but I cannot find a statement about the GSL ice cover. Admittedly, it is rather late to be asking this question but is there any possibility that the far side of the NBB ice floe is affecting the breakup by creating a standing oscillation under the floe? (A similar question can be asked for the GSL experiment but I don't have the details and the question may be irrelevant.

33. Line 390. The authors write 'Based on the previous discussion, we use the mass loading dispersion relation when the wave propagates in unfractured ice.' So, given my comment 31 above in regard to Fig. 7 of Squire (1993) can the authors be sure that it is actually the mass-loading model or could it be a flexible thin plate with a lower value of $Y^*$ or $h$? For example, page 225 of Squire (1993) states 'Fig. 7 shows that the choice of $Y^*$ is critical in determining the relative magnitudes of the wave numbers in ice and water which, since the elastic plate analogy is a parameterization, suggests caution.' Because the mass-loading model has been dismissed so many times in the wave-ice interaction literature, even for propagation in frazil ice, I am afraid I favor a reduced flexural rigidity hypothesis articulated via $Y^*$ or $h$.

34. Line 391. 'It is unclear whether the same relation applies to fragmented ice or if it rather follows the deep water relation'. It is obvious that fractured ice will not have the same dispersion relation as a continuous ice plate if the dispersion relation is based upon a Kirchhoff-Love plate (Euler-Bernoulli beam), but now one enters the murky realm of parameterization. Essentially one could envision a reduction in $Y^*$ when the ice breaks up. Or one could imagine a slightly more extreme version where after breakup $Y^* \to 0$, i.e. the medium becomes a mass-loading medium. If the ice floe starts out behaving as a mass-loading medium then physically I guess one could argue that the masses somehow change … but they don't as far as I know!

35. Line 395. I recommend adding 'open-water' as it is slightly unclear as it stands; so 'The largest open-water wave was slightly less than one meter high.'

36. Line 414. I am bothered by 'Even though the ice is thicker in the NBB experiment, the attenuation is almost one order of magnitude weaker than the attenuation in the GSL.' This actually suggests to me that Eq. (17) is producing the wrong $a_b$, i.e. either $Y^*$ or $h$ is incorrect (which would also produce an inaccurate $k^2$).

37. Line 420. There is no doubt that inelastic effects could cause a difference in attenuation between the two experiments because, as I understand it, the physical properties of the sea ice and hence its material properties were different between the two experiments. (There is no need to say anelastic, as noted elsewhere, as inelasticity covers a multitude of sins.) However, given what I have said above and the lack of amplitude measurement in the ice cover, perhaps any statements about inelastic constitutive laws are best avoided.

38. Conclusions. See the general comments section.

Technical corrections

1. The authors declare that they provide the first in situ observations of floe size distributions (FSD) resulting from wave-induced sea ice breakup. Being a pedant, I would aver that the statement is misleading as there are numerous observations of the FSD arising as a result of wave-induced sea ice breakup but that the observations have not segued into any interpretation which improves our understanding of the process. What the authors actually mean is that they are the first to have computed the FSD from observational data of wave-induced breakup. While I actually also find this statement hard to believe, I shall take the authors at their word and assume that they have perused Russian, Japanese and Chinese, as well as North American and European sea ice corpora. However, I also caution that if they haven't scrutinized the international scientific literature carefully, they should avoid making such a strong statement to avoid a rebuttal.

2. There are various instances of poor English grammar, e.g. singular words that should be plural, poor syntax where the subject is missing, weak or confusing sentence structure, a missing (adjective or adverb) article 'the', etc. In lines 116–118, for instance, '… while still allowing to study break-up in the natural environment. Such a setup also allows to have no constraint on the location of deployments and to search for the right sea ice to break.' Or 'The error on its vertical and horizontal position are respectively of 0.5 and 1.5 m.' Blunders are scattered throughout the text and need to be copy-edited out by somebody as there are quite a few indiscretions, assuming that *The Cryosphere* expects sound English prose.

3. The authors use the word 'further' consistently. In many cases, although not all, they actually mean 'farther'. Such occurrences should be corrected.

4. There is a mixture of US and English spelling, e.g. 'traveled' yet 'modelled'.

5. Line 65. 'They rather let it evolve …' should be 'Rather they let it evolve …'

6. Figure 4 caption should read Matlab not Maltab.

7. Line 154. 'consists in a series of steps' should be 'consists of a series of steps' or 'proceeds in a series of steps'.

8. Line 161–162. 'We refer the reader to (Zhang and Skjetne, 2018)' should be cited as 'Zhang and Skjetne (2018)'.

9. Line 220. 'let's compute' is a little informal.

10. Line 278. The brackets in the citation are of the wrong type, i.e. it should be Fox and Squire (1991).

11. Line 310. '… lies on the same mathematical premises?' is poor English. Replace with '… based upon the same mathematical premises?'

---

## Referee Comment (RC2)

**WIFF1.0: A hybrid machine-learning-based parameterization of**
**Wave-Induced sea-ice Floe Fracture**
**Horvat & Roach (2021)**

**General comments**
- This an interesting and well-written paper although I would recommend revisions (somewhere between minor and major) before publication.
- It makes a lot of sense to use a NN to update the FSD depending on the wave field/ice conditions etc, although maybe it should be more flexible (eg having the breaking threshold as an input parameter to the NN).
- One question is if a NN classifier is the best thing to use or can a simpler criterion be applied (eg a threshold in the variance in the strain) that would be
    i. simpler and faster
    ii. physical (as opposed to a black box). I would also mention the paper of Voermans et al (2020) (see https://tc.copernicus.org/articles/14/4265/2020/) who seemed to find such a threshold empirically by collating several observations of sea ice break-up in the presence of waves. This paper is conspicuous in its absence from the bibliography by the way.
    iii. flexible as opposed to being fixed during the training.

**Specific comments**
- P2 l50: Training on model output is a good idea for generating a variety of input wave spectra, but there is potentially other confounding factors – more details could be provided – hopefully it is still only the input and output of the SP-WIFF that is used for training?
- P2 L50: A year should be long enough to give enough variety in conditions – eg from regional, seasonal  differences.
- P4 l95: Perhaps add the RELU acronym for the activation functions as probably not many people know what it stands for (including me)
- P3: S1, S2: why does the algorithm need to converge when there is no ice-to-wave feedback?
- P4 l101, l107-108: you talk about SIC and SIT histograms, but you only input the mean SIC and SIT to the NN? Is the model run with a joint thickness/size distribution or is there just one FSD for all thickness categories?
- Fig 1
    - add "run rate" definition to caption? (I see it is defined later in the text, but it took a while to find it). Can you also define "false positive/negative rate"?
- P6 eq 5: was there a reason for wanting to weight errors in the bins for higher floe sizes more than the bins for smaller ones? Otherwise a simple RMSE might be enough. Another possible metric for comparing PDFs are the Kolmogorov-Smirnov test (this may not be differentiable, but could be used in evaluation). The SSE you've defined is reasonable though.
- Discussion: perhaps an example that could be quite pertinent to the current paper is to speed up calculation of source terms in WW3. In addition, the SP-WIFF could be enhanced by allowing some ice-to-wave feedback and could possibly output a wave source term as well as the FSD.
- For some setups (eg if an interface like OASIS-MCT were used in the wave-ice coupling), using NN-WIFF as provided would require the full wave (frequency) spectrum to be passed to the ice model which is quite costly. Perhaps some kind of dimensionality reduction could be performed to reduce the parameters that had to be passed? This could possibly be done with a NN as well, in order to fit in with the current structure.
- Fig 5, bottom row: 2 orders of magnitude difference in 4.5% of the total sea ice area is not so small; it becomes more significant as a fraction of the MIZ. Since (as you note yourselves) this is on the MIZ-pack boundary it could be worth another look at the classification – I think it is worth another trial run with the classifier replaced by a simpler and perhaps more stable criterion.

**Typos**
- P2 l5: "it overall computation times by an order of magnitude" to "it increases overall computation..."
- p3 l61: equation should possibly be something like this

$$S(\lambda) = \int_0^{2\pi} S(\lambda, \theta) d\theta$$

?

- P11:Integrating these simulations *using on* the Cheyenne supercomputer
- p11: "NN-WIFF reduces the overhead associated with simulating wave-ice fracture without significant added computational cost" to " NN-WIFF reduces the overhead associated with simulating wave-ice fracture"?
- p12 l61-62: "Because of the ease of obtaining training data from climate model output, this parameterization acceleration approach has, and could continue to, find" - "… has found,… to find,"
- p12 l266: paramterization,

---

## Referee Comment (RC3)

**Review of "Aerial observations of sea ice break-up by ship waves"**
**by Dumas-Lefebvre & Dumont (2021)**

This paper is quite a good one and is one of the first to capture a break-up event in the field that is solely caused by waves. I'd recommend minor revisions.
I have some specific comments below.
Regards,
Timothy Williams

**Specific comments**

- p2 "*Floe size is also important for constraining wave propagation and attenuation...*" I would make it clearer the differences between parametrisations depending on FSD and physics which is still largely unknown.

- P3 "*By assuming that ice breaks up where the deformation is maximal, Roach et al. (2018) obtained that the fracture of sea ice by waves leads to a preferential size. This means that wave-induced break-up leads to a bell-shaped FSD, a result that indicates that the morphology of floes resulting from breakup might not be well represented by a power law*". The bell shape in this case probably corresponds to peaks in the wavelength spectrum since there are no hydrodynamics in their model (they create a surface profile from the open water spectrum)

- p3. Mokus & Montiel (2021) could be worth discussing – they produced log-normal FSD from hydrodynamical simulations.

- p4. "*Indeed it is possible to study wave-ice interactions in laboratories as Herman et al. (2018) did, but it is not clear if the results directly apply to the natural environment due to the difference of scale and properties between laboratory-grown ice and sea ice.*" It should be clarified that the current experiment is not completely full-scale as the waves from the ship were very short compared to "natural" waves.

- figs 8-10: captions don't say which expt is which

- p13: the area-weighted PDF does indeed seem more representative and also has a convenient corresponce to FSD formulations in models like  in Roach et al (2018)

- p15: it was useful to have this information about the origin of x* here. There was a mixture of beam and thin plate here though – moment of inertia for a beam is [width] [thickness]^3/12 and no Poisson's ratio; for a plate EI is swapped for the flexural rigidity Eh^3/(12(1-nu^2)).

- P16. Good point about the half-wavelength and x* lengths corresponding to maxima in deformation. Here could be a good point to mention Asplin (2012), who noticed breaking into strips of half-wavelength in a place far from the ice edge. As in the Mellor quote, it seems like the presence of the ice edge is quite important, that the break-up occurs so fast (after very few cycles) that the fracture always seems to occur at the closest maxima to the edge which you and Herman et al (2018) found to be correlated to the x* length. Another paper which could be relevant is Williams and Squire (2014) who looked at results from a hydrodynamic model to see that maxima in long floes were separated by half a wavelength (more like the Asplin case), but they didn't look at the distance from the ice edge to the first maximum.

- P17. Not totally convinced of the importance of fatigue since it sounds like the break-up front is advancing very fast. Maybe it is important at the end of the MIZ – perhaps there is more spread in floe size there?

- P18-19: "*which is close to ½, the ratio … in deep water*" should maybe change to "*for deep water without any ice cover*" The thing to look at would be the change in wavelength going from open water to choose the dispersion relation and then calculate the group velocity, rather than assuming the break-up front advances at the group velocity. The work of Sakai & Hanai (2002) would be relevant too, who showed a transition from elastic plate to mass loading behaviour as floe length decreased (with artificial floes in a laboratory) so fragmented ice behaving in a mass loading way is consistent with this. It would be an interesting result though if $c_b$ and $c_g$ were about the same, and would make some sense as well.

- Eqn (16): maybe a transmission coefficient should be multiplied by $a_0$ to get the amplitude in the ice? This would be smaller for thicker ice, making the 2$^{nd}$ attenuation coefficient even smaller compared to the 1$^{st}$. The difference is indeed marked between the 2 cases. Another counterintuitive thing is that the thin ice is broken into smaller floes which would scatter less and would be expected to have lower attenuation than the longer floes. Other FSD-dependent parameterisations like creep also attenuate waves less when the floe size is lower. Perhaps there is more friction between floes or something like that (bigger perimeter), but like you say that is a bit out-of-scope.

- p22: "*The modal shape of the FSDs informs us that sea ice breaks up systematically at strains lower than the extrema such that material fatigue is of important when considering breakup (Langhorne et al., 1998)*". I don't follow this argument – it shows that there is a preferential length scale doesn't it? The spread around the mode could come from many sources (as you say in the next sentence a bit) – ice heterogeneity (as you mention), an irregular ice edge, waves from a spread of angles. Ship waves are curved also – this could maybe have an effect over a longer distance into the ice.

**typos**

- Fig 4 caption: maltab → matlab

**References**

Asplin, M. G., Galley, R., Barber, D. G., and Prinsenberg, S. (2012), Fracture of summer perennial sea ice by ocean swell as a result of Arctic storms, *J. Geophys. Res.*, 117, C06025, doi:10.1029/2011JC007221.

Mokus, N. G. A., & Montiel, F. (2021). Wave-triggered breakup in the marginal ice zone generates lognormal floe size distributions. *The Cryosphere Discussions*, 1-33.

Sakai, S., & Hanai, K. (2002, December). Empirical formula of dispersion relation of waves in sea ice. In *Ice in the environment: Proceedings of the 16th IAHR International Symposium on Ice* (pp. 327-335).

Williams, T. D. and Squire, V. A. (2014). Wave-induced strains in ice floes. *22$^{nd}$ IAHR International Symposium on Ice*: Singapore, p. 814-821

---

## Author Comment (AC2)

**Review of "Aerial observations of sea ice break-up by ship waves" by Dumas-Lefebvre & Dumont (2021)**

Comment

> This paper is quite a good one and is one of the first to capture a break-up event in the field that is solely caused by waves. I'd recommend minor revisions. I have some specific comments below. Regards, Timothy Williams

Answer

> Thank you for your consideration of our manuscript. We are pleased that someone who has a lot of experience in the specific field of wave-induced sea ice breakup reviews our article. Your comments helped us clarify a certain aspects of the paper, and brought our understanding of the problem further. Please see below for the answers to your specific comments.

**Specific comments**

Comment

> p2 "Floe size is also important for constraining wave propagation and attenuation..." I would make it clearer the differences between parametrisations depending on FSD and physics which is still largely unknown.

Answer

> Right. We modified the sentence to "Floe size is also important for constraining some parameterisations of wave propagation and attenuation, although it is still unclear whether and how is it significant in reality."

Comment

> P3 "By assuming that ice breaks up where the deformation is maximal, Roach et al. (2018) obtained that the fracture of sea ice by waves leads to a preferential size. This means that wave-induced break-up leads to a bell-shaped FSD, a result that indicates that the morphology of floes resulting from breakup might not be well represented by a power law". The bell shape in this case probably corresponds to peaks in the wavelength spectrum since there are no hydrodynamics in their model (they create a surface profile from the open water spectrum)

Answer

> I understand your point but we wanted to highlight the fact that by considering the breakup idealization of (Dumont et al., 2011), i.e. sea ice conforms to wave profile and break at strain extrema, they obtain a FSD which is different than a power law truncated at $\lambda/2$.

Comment

> p3. Mokus & Montiel (2021) could be worth discussing – they produced log-normal FSD from hydrodynamical simulations.

**Answer**

We have added the following text between lines 76 and 77 of the manuscript: "More recently, Mokus & Montiel (2021) created a 2-D hydrodynamic model for wave-induced sea ice breakup which combines linear wave theory and viscoelastic sea ice rheology in order to compute the scattering of wave by sea ice floes. Using an empirical strain threshold to define the floe size resulting from breakup, they obtained that the FSD follows a lognormal distribution under realistic wave forcings thus demonstrating that a preferential size is indeed generated by the process. They also show that the median floe size evolves with both wave period and ice thickness, result that partly contrasts with the findings of Fox & Squire (1991) and Herman (2017) in which the FSD is independent of the sea state."

**Comment**

p4. "Indeed it is possible to study wave-ice interactions in laboratories as Herman et al. (2018) did, but it is not clear if the results directly apply to the natural environment due to the difference of scale and properties between laboratory-grown ice and sea ice." It should be clarified that the current experiment is not completely full-scale as the waves from the ship were very short compared to "natural" waves.

**Answer**

The text will be modified to better reflect the fact that ship generated waves are indeed different from wind-generated waves. "Indeed it is possible to study wave-ice interactions in the laboratory, as Herman et al. (2018) did, but it is not clear if the results directly apply to the natural environment owing mostly to the complex life history of naturally-grown sea ice compared to the more homogeneous growth conditions of the laboratory. Even though ship-generated waves are different from wind-generated waves, their period and amplitude are nonetheless representative of natural waves generated in short fetch seas that impacting ice conditions."

**Comment**

figs 8-10: captions don't say which expt is which

**Answer**

Corrected

**Comment**

p13: the area-weighted PDF does indeed seem more representative and also has a convenient correspondance to FSD formulations in models like in Roach et al (2018)

**Answer**

Glad to hear your approval about the method we propose.

**Comment**

p15: it was useful to have this information about the origin of x* here. There was a mixture of beam and thin plate here though – moment of inertia for a beam is width$*h^3/12$ and no Poisson's ratio; for a plate EI is swapped for the flexural rigidity $Eh^3/(12(1-\nu^2))$.

Answer

Thank you for pointing out this inconsistency. We found important to present again the derivation of Mellor (1983) and Hetenyi (1946) in order to explicitly clarify and error pointed out and corrected by Boutin et al. (2018). As Boutin et al. (2018), we want to use the flexural rigidity of a plate, instead of a beam. We changed the presentation of the derivation and made it clear when we stop considering a beam and consider a plate instead. Here is the proposed text:

"This implies that the location of the maximum bending moment, and therefore of maximal deformation, is $x^* = \frac{\pi}{4}\left(\frac{4EI}{k_f}\right)^{1/4}$ where $k_f = \rho_w g$. Even though it is derived for a beam, the same logic applies to a plate, which is a better representation of an ice floe. For a plate, $EI = \frac{Yh^3}{12(1-\nu^2)}$, with $Y$ the Young's modulus for sea ice, $h$ the plate thickness, $\nu = 0.3$ the Poisson ratio, $\rho_w \simeq 1025$ kg m$^{-3}$ the sea water density and $g$ the gravitational acceleration."

Comment

P16. Good point about the half-wavelength and x* lengths corresponding to maxima in deformation. Here could be a good point to mention Asplin (2012), who noticed breaking into strips of half-wavelength in a place far from the ice edge. As in the Mellor quote, it seems like the presence of the ice edge is quite important, that the break-up occurs so fast (after very few cycles) that the fracture always seems to occur at the closest maxima to the edge which you and Herman et al (2018) found to be correlated to the x* length. Another paper which could be relevant is Williams and Squire (2014) who looked at results from a hydrodynamic model to see that maxima in long floes were separated by half a wavelength (more like the Asplin case), but they didn't look at the distance from the ice edge to the first maximum.

This is a very good remark. In our opinion, the event that [Asplin et al., 2012] describe does not provide sufficiently detailed information about the floe size (only a few visual observations are mentioned), and wavelength (they use the deep open water dispersion relation) that would allow a solid relationship to be found between the two. Moreover, they infer that the FSD follows a power law instead of measuring it. However, you are right about how we should interpret $x^\star$ and $\lambda/2$ and we will modify our discussion accordingly, adopting the following general line of thought. If an observed FSD is bounded by $x^\star$, this would highlight that the fracture is tied to the ice edge, while if it's bounded by $\lambda/2$, it would mean that waves have had the chance to propagate further into the ice sheet before breaking it up, in other words that break-up happened more slowly with respect to the waves. What our results seem to suggest is that break-up occurred as soon as the strain reached the critical value, at a location that is a certain distance from the ice edge where the strain is zero.

Answer

P17. Not totally convinced of the importance of fatigue since it sounds like the break-up front is advancing very fast. Maybe it is important at the end of the MIZ – perhaps there is more spread in floe size there?

Comment

Answer

You are right. What we wanted to discuss here is that the location where the ice breaks is smaller than $x^*$, which might not be related to fatigue but instead to the fact that the maximum strain is larger than the critical strain. Fatigue, as you say, will play a role if an unbroken ice plate has been deformed by the passing of many waves. There is evidence of this in our observations. The discussion will be modified accordingly.

Comment

P18-19: "which is close to $\frac{1}{2}$, the ratio ... in deep water" should maybe change to "for deep water without any ice cover" The thing to look at would be the change in wavelength going from open water to choose the dispersion relation and then calculate the group velocity, rather than assuming the break-up front advances at the group velocity. The work of Sakai & Hanai (2002) would be relevant too, who showed a transition from elastic plate to mass loading behaviour as floe length decreased (with artificial floes in a laboratory) so fragmented ice behaving in a mass loading way is consistent with this. It would be an interesting result though if $c_b$ and $c_g$ were about the same, and would make some sense as well.

Answer

This will be corrected. However, here we do not assume that the break-up front advances at the group speed. We measure the speed of the break-up front, named $c_b$, and then compare this value with the wave group speed $c_g$, which is estimated the observed in-ice wavelength and period, before the ice breaks into smaller floes. We were not able to reliably measure the period and wavelength in fragmented sea ice. $c_g$ is then compared to the theoretical values assuming mass loading and flexural dispersion relation, from which we conclude that the former applies better to our problem. This will be clarified in the manuscript.

Comment

Eqn (16): maybe a transmission coefficient should be multiplied by $a_0$ to get the amplitude in the ice? This would be smaller for thicker ice, making the 2nd attenuation coefficient even smaller compared to the 1st. The difference is indeed marked between the 2 cases. Another counterintuitive thing is that the thin ice is broken into smaller floes which would scatter less and would be expected to have lower attenuation than the longer floes. Other FSD-dependent parameterisations like creep also attenuate waves less when the floe size is lower. Perhaps there is more friction between floes or something like that (bigger perimeter), but like you say that is a bit out-of-scope.

Answer

We are delighted by the discussion triggered by the counterintuitiveness of our results. We adopted a point of view where we do not assume any underlying attenuation mechanism. Assuming a transmission coefficient would mean that scattering happens, which is probably the case. However, we want to let the door fully opened as to whether other mechanisms (known or unknown) are at play, only to conclude that, like Boutin et al. (2018) says, there are multiple possible attenuation mechanisms and they still need to be further investigated.

 p22: "The modal shape of the FSDs informs us that sea ice breaks up systematically at strains lower than the extrema such that material fatigue is of important when considering breakup (Langhorne et al., 1998)". I don't follow this argument – it shows that there is a preferential length scale doesn't it? The spread around the mode could come from many sources (as you say in the next sentence a bit) – ice heterogeneity (as you mention), an irregular ice edge, waves from a spread of angles. Ship waves are curved also – this could maybe have an effect over a longer distance into the ice.

Answer The modal shape of the FSD indeed tells us that there is a preferential size, so would have done a linearly increasing distribution bounded by $x^*$. In the the latter case, where the floe size would have primarily been of $x^*$ with decreasing probability towards small floes, $x^*$ would have been the preferential size and thus the ice would have mainly broke where strain is maximal. On the contrary, observing a preferential size at a value lower than $x^*$ indicates that the ice breaks at a critical strain that is lower than the maximal strain.

**Typos**

Comment Fig 4 caption: maltab → matlab

Answer Corrected

**References**

[Asplin et al., 2012] Asplin, M. G., Galley, R., Barber, D. G., and Prinsenberg, S. (2012). Fracture of summer perennial sea ice by ocean swell as a result of Arctic storms. *Journal of Geophysical Research: Oceans*, 117(C6).

---

## Author Comment (AC3)

**Aerial observations of sea ice break-up by ship waves**
**by Elie Dumas-Lefebvre & Dany Dumont**

**General comment**

Comment

This is an enjoyable little paper which reports some unique results on a topic of meaningful contemporary interest that has been reinvigorated by the effects of global climate change on the polar and subpolar seas. The work is reported clearly and is mostly well-written. Although the observations and subsequent analysis are limited in extent, they are interesting and thought-provoking, as they intersect earlier theoretical conjectures about the way sea ice fractures under the action of ocean waves – namely in regard crack separation and the FSD configuration that is created as a consequence. Nonetheless, sometimes assertions are made in the paper that are not adequately justified by evidence. The authors should be careful of this and ensure that what they are saying is correct and that earlier papers are cited where applicable as conclusions have been reached in some cases without the benefit of results reported in a previously published paper that remains uncited. This notwithstanding, and appreciating that keeping track of every publication in a research field is impossible, in the reviewer's opinion a revision of the work to accommodate the specific comments and technical corrections below will improve the paper sufficiently for it to be published in Cryosphere. I am aware that my review will seem pernickety and possibly irksome at first sight but my intention has been to finesse the paper because I believe its results should ultimately be disseminated through publication. In sum, I feel that the authors are being too assertive in regard to the outcomes of their study. The methods they have used are novel, exciting and have tremendous potential. I congratulate them for collecting and compiling a manuscript to publish these observations. I am inclined to disagree with some of their unequivocal affirmations which, to me, are not always based on robust evidence and perhaps to some extent show a lack of deep understanding of this multifaceted research topic. I recommend that the manuscript should be published but only after the authors have acted on the comments below.

Answer

Thank you for considering our manuscript. At first, it has been a challenge to open our minds to the perspective you brought up in your review but, in the end, there is a clear benefit to include the ideas and suggestions presented in your comments to our paper. Key aspects of the review made us realize that the experiment we carried out are not representative of how wave-induced sea ice breakup universally behaves in the natural environment. Furthermore, even though $x^*$ has been used in numerous studies, using this quantity derived from a quasi-static approach in order to understand the underlying physics of a inherently dynamic problem might be wrong. Quantities derived from scattering models might be better tools for that sake since they consider the dynamics of breakup. In the end, your review has brought our article at an higher level but has also helped us to broaden our theoretical comprehension of wave-induced sea ice breakup. For the answers to your specific comments and technical corrections, please see below.

**Specific comments**

**Comment**

Line 3–4. What does 'When represented as probability density functions weighted by the surface of ice floes' mean? That is, how can something be weighted by a surface?

**Answer**

Floe size distributions have traditionally been represented by frequency distributions, i.e. as if all floes are non-dimensional objects that have the same probability, irrespective of the surface they occupy in the space they live in. However, sea ice floes are arranged on a surface without overlapping such that the total area covered by floes can't be larger than the total area that is considered. The area-weighted distribution wishes to characterize *the probability of finding a floe of size d at a randomly chosen position in the image.* This definition is similar to the so-called ice thickness distribution (ITD) that is used in sea ice models. That being said, we acknowledge that labelling this approach with the words "area-based" or "area-weighted" can be confusing. Instead, similarly to the vocabulary used in the ice modeling community for the ITD, we propose using *partial concentrations*, which refers to the portion of the total surface that is covered by floes of a given size, rather than "area-weighted" or "area-based". Eq. (3) of the manuscript shows the mathematical definition of this FSD.

**Comment**

Line 9 the 'mass loading dispersion relation' won't be known to many readers. I am assuming a citation is not allowed in the abstract, so a definition or explanation should be provided here.

**Answer**

This will be changed in the abstract.

**Comment**

Line 9–10. 'Moreover, our experiments show that thicker ice can attenuate wave less than thinner ice'. This is counterintuitive so, if the authors still believe this to be the case after reading my comments, I would suggest adding a short statement explaining why, e.g. Moreover, our experiments show that thicker ice can attenuate waves less than thinner ice, because . . . '. Notice also that 'wave' has been altered to 'waves'.

**Answer**

Yes we will better justify that statement, if it is still a valid conclusion considering the changes to be expected with lower values of $Y^*$.

**Comment**

Line 18. What do the authors mean by 'larger' waves? Waves of greater amplitude or waves of longer wavelength, or both?

**Answer**

Good point, the text of line 18 has been changed to:
"... that generate a more energetic wave field in the Arctic basin ... "

**Comment**

Line 20. Why do waves 'change the mechanical properties of the ice'? Are the authors hinting at fatigue or are they using the word 'ice' loosely, meaning the ice-cover as a whole?

**Answer**

What we want to highlight in line 20 is the fact that the broken up sea ice does not have the same rheology, and hence large scale mechanical properties, than a consolidated ice pack. This phrase says that wave-induced breakup changes mechanical properties of sea ice by altering the scale of the floes, not that wave directly impact the mechanical properties of the floes. From your comment, we understand that the term "mechanical" is not useful here and the idea is adequately transmitted through the word "dynamics". We will thus remove the former.
The text will be modified to :
"The increasingly energetic waves may then have greater potential to break up sea ice thus generating a larger MIZ so that, on the large scale, changes in the dynamics of the ice cover and in ocean-atmosphere heat exchanges could be observed."

**Comment**

Line 34. Don't the usual definitions of anelastic and inelastic designate that anelasticity is a particular type of inelasticity; if so why do the authors list them separately suggesting that they represent independent material behaviors.

**Answer**

We distinguished the two because they are treated separately by certain authors, as reviewed for example by Boutin et al. (2018). Since we do not intend here to provide any explanation or validate any theory here, we are inclined to keep that discussion focused specifically on what could generate the counterintuitive results we obtain, namely that thicker ice attenuate waves less (see comment above).

**Comment**

Line 36. I find it hard to believe that 'there are no observations that directly relate the FSD to a given process in the natural environment', but accept what the authors intended in the sentences that follow in the text.

**Answer**

It is true that it may be a bold statement since some studies may not have been included in our literature review since we cannot truly know every studies that have been published. We will change this sentence to :
'To our knowledge, there are no observational studies that directly relate the FSD to the processes that generated it in the natural environment.'

**Comment**

Line 67. 'This means that wave-induced break-up leads to a bell-shaped FSD, a result that indicates that the morphology of floes resulting from breakup might not be well represented by a power law.' Didn't Montiel, F. and Squire, V. A. Modelling wave-induced sea ice break-up in the marginal ice zone. Proceedings of the Royal Society of London, Series A 473(2206): 1–25 (2017) deduce a similar result? This paper should be cited incidentally, as it concerns breakup by waves.

**Answer**

Right. The following sentence will be added : "Montiel & Squire (2017) also found out that either a modal or bimodal distribution was generated from wave-induced sea ice breakup, thus suggesting that the observed power law does not come out from this process alone."

**Comment**

Line 70. I personally think the paper Fox, C. and Squire, V.A. On the oblique reflexion and transmission of ocean waves from shore fast sea ice. Philosophical Transactions of the Royal Society A 347(1682): 185–218 (1994) is a better one to cite than Fox and Squire (1991), as it is more general and includes everything reported in the 1991 paper, but I accept that the authors are quoting a specific statement.

**Answer**

Thank you for your understanding and for the suggestion.

**Comment**

Line 87. The author state 'quantitative analysis is required to fully test these hypotheses' but isn't it obvious that, all things being equal and ignoring any small amplification arising due to the free edge boundary condition, long-crested propagating waves stress all parts of an ice plate similarly, so the specific wavelengths involved will be less important unless standing waves are created.

**Answer**

Indeed, the use of the words "quantitative analysis" alone is not adequate since both observational and modeling studies can be quantitative. The words "quantitative analysis of observational data" will thus be used for clarity.

**Comment**

Lines 100–108 are the nub of the paper yet I find the paragraph poorly expressed and somewhat blah. As the authors rightly say, it is incredibly difficult to get data on naturally-occurring wave-induced breakup of ice floes because it demands the stars to be aligned even when instrumentation is available. What this particular study does is to create the waves and then measure the outcome. The only thing missing is a measurement of the induced curvature or stress. Wouldn't that be nice! At minimum break the paragraph at line 104, i.e. start a new paragraph at 'In this study, we use . . . ' so the reader can immediately see the goals and any limitations of the paper.

Answer

This paragraph has been changed to the following :
"Few observational studies about natural break-up have been made yet mainly because the MIZ is an arduous area to sample directly from. It is indeed hard to be in the MIZ at the right place and at the right time, with good but not too harsh weather conditions for break-up to happen, and with the right apparatus to measure all relevant variables during a natural break-up event.
Rather than waiting for the stars to be aligned in the natural environment, we chose to create waves with a ship in order to simulate breakup events. With the help of an unmanned aerial vehicle (UAV or drone) and image processing, the breakup experiments conducted in the Gulf of Saint-Lawrence (GSL) and in the northern Baffin Bay (NBB) allowed us to measure the outcome of small period waves breaking naturally grown sea ice. While no apparatus to measure the strain of the ice or incident wave properties were successfully deployed, it was nonetheless possible to infer some key wave and ice properties and to fully characterize the FSD, the breakup speed and its extent. When compared to thin elastic plate theory, these results give insight on the underlying physics of wave-induced sea ice breakup."

Comment

Line 110–111. What does the word localized mean in 'First, a large level ice floe having a side exposed to open water is localized'?

Answer

Good point. The use of localized here is derived from french and a better word for this sentence would be identified. The sentence will now read : "First, a large level ice floe having a side exposed to open water is identified"

Comment

Line 116. In'... hands a better management of weather conditions ...', I do not know what 'hands' means in this context. Is this a technical word? Or is it bad English and means 'provides'.

Answer

It might be bad english. Thank you for providing us an alternative word for this sentence.

Comment

On Line 127 the authors state that the sea ice in the Gulf of St. Lawrence was grey and grey-white between 10 and 30 cm-thick'. This is an important observation that the reviewer will refer to later. No sensor was deployed on the ice itself.

Answer

It is for sure not as precise as a in situ ice thickness measurement but it is still a measure that has been made by an officer of the Canadian Ice Service. It would have been amazing to have data on the ITD and the wave spectrum as well as the FSD but since the experiment was opportunistically planned on the ship, we could not have such data.

Comment Likewise lines 140–142 indicate that sea ice in Northern Baffin Bay during the experiment there was heavily rotted first year ice between about 40 and 60 cm. Two SKIb wave buoys deployed on the ice were capsized, unfortunately, so provided no data.

Answer Yes.

Comment Line 205. I note for future reference that the waves are relatively short period, as expected for a Kelvin wake, namely around 4 s.

Answer Good.

Comment Figures 5–8 are awesome!

Answer Thanks ! If you want to see the video footage of the GSL breakup, it is accessible here : `https://www.researchgate.net/publication/356568634_Aerial_footage_of_wave-induced_sea_ice_breakup_in_the_Gulf_of_Saint-Lawrence/stats`

Comment Line 255–258. I agree with the opinions expressed in this paragraph but the authors must also remind the reader here that the waves being discussed here are of modest period, i.e. around 4 s. Indeed, in regard to natural ocean waves they would be categorized as 'chop'. Furthermore, the sea ice in the experiment is somewhat distinctive, i.e. a subcategory of what is encountered in the polar oceans. It would be disingenuous to suggest that the data being presented are universally applicable to all wave-ice interaction scenarios. This is not intended to lessen the importance of the results being presented, which I regard highly, but rather to ensure that the facts are presented accurately.

Answer With your comment and the one of the other reviewer, we have added information about ship wave at line 103 with the following sentence :
"Indeed it is possible to study wave-ice interactions in the laboratory, as Herman et al. (2018) did, but it is not clear if the results directly apply to the natural environment owing mostly to the complex life history of naturally-grown sea ice compared to the more homogeneous growth conditions of the laboratory. Even though ship-generated waves are different from wind-generated waves, their period and amplitude are nonetheless representative of natural waves generated in short fetch seas that impacting ice conditions."
It is true that our result do not apply universally to every type of ice and to every type of waves, thank you for underlining that detail.we will consider that when rewriting our discussion.

**Comment**

Eq. (4) assumes the ice floe is an Euler-Bernoulli beam, i.e. the sea ice is thin and homogeneous through its thickness. This should be stated, as sea ice in nature has a temperature and brine volume gradient that renders the latter assumption an approximation.

**Answer**

This will be added in the text.

**Comment**

While the statement in lines 278-283 is correct, I do wonder whether the small local peak that occurs near the ice edge is sufficient to explain why the ice progressively fractures from its margin to its interior, given the degree of approximation inherent in the hydroelastic model.

**Answer**

See our response to the comment below.

**Comment**

Line 283. I am not sure what 'Unfortunately, there is no simple analytical solution we can use to scale our result' is saying. There are a wealth of hydroelastic studies of wave-ice interaction dating back to papers published done in the 1950s and the first Weiner-Hopf analysis of Evans and Davies in 1968 which was generalized in Williams, T.D. and Squire, V.A. Scattering of flexural–gravity waves at the boundaries between three floating sheets with applications. Journal of Fluid Mechanics 569: 113–140 (2006). It is also worth noting that by ignoring the mass of the plate, a fully algebraic result was obtained in Tkacheva, L.A. Scattering of surface waves by the edge of a floating elastic plate. Journal of Applied Mechanics and Technical Physics 42(4): 638–646 (2001), which could presumably be used. It is disappointing to me that a more authoritative model has not been used when so much theory is available; the fragility of the wave-ice interaction topic generally is not situated in mathematical theory but in the paucity of data to validate theory. On the other hand, I am not suggesting that the authors can ameliorate this problem at this stage by reworking their analysis, I am simply conjecturing that a different approach could have changed the manuscript from where I began this review, i.e. 'This is an enjoyable little paper', to a something more substantive.

**Answer**

This work is the main outcome of a Master's research project and you are right that a thorough literature review of hydroelastic studies was not carried out, one of the many aspects of the observational work forming the central part of the manuscript. Nonetheless, we are very interested to deepen the discussion of our results towards. We thus thank you for suggesting the article of L.A. Tkacheva. We will definitely consider this paper and references therein in our discussion. From comments received from two reviewers, we decided to emphasize the significance of the observation of a progressive break-up from the ice edge, which is an original contribution of this paper, without claiming too strongly that we know what is going on.

Comment

Line 284. I will point out the obvious. The Hétenyi (1946) model has no fluid dynamics, which I would perceive as fundamental to interpreting an ocean wave phenomenon.

Answer

We agree that the use of a value derived from the theory of Hétenyi (1946) might not be adequate in order to obtain information about the contribution of sea ice properties in determining floe size resulting from breakup. Nonetheless, it has been used in many studies since 2011 (e.g. Toyota et al; 2011, Williams et al.; 2013a-b, Hermann et al., 2017; Boutin et al.; 2018) so that we deemed necessary to compare it to our data and to show how $x^*$ was derived in order to clarify this topic in the scientific community. In the revised version of the manuscript, we will specify that and use Hetenyi's formulation as a material-dependent length scale that represents in essence our strain develops in a semi-infinite plate. We will also better link with studies that explicitly consider waves.

Comment

In Eq. (5), the authors need to say that it is an Euler-Bernoulli beam that they are using as a model. Strictly, they would actually be better to use a Kirchhoff–Love plate, i.e. the plate equivalent of an Euler-Bernoulli beam, which would introduce a Poisson ratio effect. However, both suffer from the obvious approximation that the sea ice is assumed to be homogeneous through its thickness. (I note that most theory assumes a similar paradigm on the basis that the waves are long compared to the thickness, so I am not criticizing the authors for assuming homogeneity, I am simply advising them to tell the reader what they are assuming.)

Answer

The paragraph preceding eq. (5) as been modified to :
"[?] considers a semi-infinite Euler-Bernoulli beam having constant thickness $h$ which extends along the $x$ axis. When submitted to a load $P$ acting downwards at its edge, a vertical deflection of the beam's edge is generated and imposes a bending moment $M$ defined as ..."

Comment

Line 288–291. This is confusing. In consecutive sentences the authors refer to bending moment, moment of inertia and moment. The third occurrence is the bending moment. Why not just say M vanishes for large x. And to be pedantic, isn't I the second moment of area rather than the moment of inertia, i.e. no mass?

Answer

The text between eq. (5) and eq. (6) has been changed to :
"where $E$ and $I$ are respectively the elastic modulus and the second moment of area of the beam, i.e. its massless inertia (Hetényi, 1946). Considering a stress-free condition at the edge and a that M vanishes for large $x$, the general solution is"

Comment

Line 294. 'First-order derivative of Eq. (6)' with respect to x.

Answer

Corrected.

Comment

Ah. Eq. (8) suddenly introduces $\nu$, which we are told later is Poisson's ration. This means that the authors are actually considering a Kirchhoff-Love plate after all, so my comment 23 is superseded. However, I think I could argue that this section needs a good tidy up to be consistent.

This section has been replaced by the following in order to clarify the mathematical approach as well as physical considerations and approximations :

"Considering a stress-free condition at the edge and a that $M$ vanishes for large values of $x$, the general solution is

$$M = -\frac{P}{\mu}e^{-\mu x}\sin\mu x, \qquad \mu = \left(\frac{k_f}{4EI}\right)^{\frac{1}{4}}, \tag{1}$$

where $k_f$ is the foundation modulus, which can be viewed as a Hooke's constant, and $x$ is the axial direction of the beam (Hetényi, 1946). Setting the first-order derivative of Eq. (1) with respect to $x$ to zero, we obtain the following algebraic equation

$$e^{-\mu x}(\cos\mu x - \sin\mu x) = 0, \tag{2}$$

which is satisfied when $x \to \infty$ or when $x = (4n+1)\pi/4\mu$ with $n = 0, 1, 2, ....$. This implies that the location of the maximum bending moment, and therefore of maximal deformation, is

$$x^* = \frac{\pi}{4}\left(\frac{4EI}{k_f}\right)^{1/4}. \tag{3}$$

Even though $x^*$ is derived for an Euler-Bernoulli beam, we insert the second moment of area $(I)$ of a Kirchoff-Love plate and use the Young's modulus $Y$ as the elastic modulus $E$ of the plate in order to use this framework in the context of the sea ice in our experiments. That way, we have

$$E = Y, \quad I = \frac{h^3}{12(1-\nu^2)} \tag{4}$$

which implies that

$$x^* = \frac{\pi}{4}\left(\frac{Y^*h^3}{3\rho_w g(1-\nu^2)}\right)^{1/4}. \tag{5}$$

with $Y$ being the Young's modulus for sea ice, $h$ the plate thickness, $\nu = 0.3$ the Poisson ratio, $\rho_w \simeq 1025$ kg m$^{-3}$ the sea water density and $g$ the gravitational acceleration. Another consideration made here is that, in order to take into account for the fact that the plate lies on water, the foundation modulus was set to $\rho_w g$."

Answer

Line 304. The authors write that 'Mellor (1983) used this framework for determining a flexure-induced fracture distance in the context of ice rafting, not for the case of wave-induced breakup.' This is important as Mellor is using his analysis for a quasi-static problem, while the authors are using it for a dynamic problem. This is why I dislike what they have done.

Comment

Answer

We understand your comment but as we said in an earlier comment, it is because $x^*$ has been used by the scientific community in various situations that we felt the need to include it in our analysis. I although think that it should be important to bring up the limitation of this framework in the discussion.

The following is added at line 310 :

"Moreover, since the framework used to obtain $x^*$ is quasi-static and wave-induced sea ice breakup is inherently dynamic, is $x^*$ the right tool to gain insight on the underlying physics of this process ?"

Comment

Line 304–310. Rather a dubious argument for all the reasons I have just articulated, namely that the theory being used is inappropriate for the dynamical data set being modeled.

Answer

Following the answer above, it is true that using the bending moment in a quasi-static manner to approach a fundamentally dynamic problem might not be the best approach. Nonetheless, since $x^*$ has been used wrongfully in some studies, we wanted to elaborate the origins of this quantity in order to rectify the assumptions made when using it. Your comment bring this rectification a step further in the sense that, even after rectifying the theory, it remains a quasi-static framework that could be inappropriate to describe wave-induced sea ice breakup. We will for sure make a point on that in our discussion. We thank you for pointing out that detail. It breaks up the paradigm a number of authors were trapped into and opens up the way for further discussion.

Comment

'The latter causes the ice to break at strains lower than its initial flexural rigidity.' How can this be? Strain is dimensionless. Flexural rigidity has units of Pa m3. This makes the sentence nonsense.

Answer

Strains will be replaced with stresses in the sentence.

Comment

I am rather worried by the arguments used starting at line 316 and 338. I simply don't believe the value of the effective modulus is nearly as high as the authors have estimated. Assuming the arithmetic is correct—and I haven't checked, the argument about the unmeasured salinity is flawed to my mind. The sea ice in question is relatively thin and warm, and the NBB floe is 'heavily rotten' according to the authors, so I would expect a much lower effective modulus. I believe the cited argument about the salinity being less later in the season relates to desalination mechanisms that are not present in warm, highly saline, sea ice of modest thickness (especially acknowledging the uncertainty around NBB thickness, apropos lines 344–349). as sea water at 30–35 ppt flushes a good proportion of the lower parts of the ice matrix. Sea ice is a mushy layer as reported by Feltham, D. L., Untersteiner, N., Wettlaufer, J. S., and Worster, M. G. Sea ice is a mushy layer, Geophysical Research Letters 33: L14501 (2006). My advice is to think through this section very carefully, as the value of effective modulus calculated is way too high in my opinion and, unfortunately, this has a bearing on the subsequent analyses and conclusions. Incidentally, while Cox and Weeks (1983) is undoubtedly the most comprehensive publication on brine volume, an easier analysis was completed earlier by Frankenstein, G. and Garner, R. Equations for determining the brine volume of sea ice from $-0.5°$ to $-22.9°$ C. Journal of Glaciology, 6(48): 943-944 (1967).

Answer

When analyzing the results, we were also worried that we might come to wrong conclusions with respect to the dispersion relation. The fact that we only have an uncertain estimation of ice thickness and that we did not measure sea ice salinity, temperature or porosity leads to large uncertainties for the elastic modulus. This is what we wanted to show with shaded areas in the figures 10 and 12. However, thanks to a thorough sensitivity analysis that includes much smaller values of $Y$, we found that our conclusions are robust to these uncertainties. We recall that we measured a decrease of the wavelength. However, flexural waves, irrespective of the value of Young's modulus are always faster and thus longer than open water waves or waves-in-ice following the mass loading. The break-up speed $c_b$ is what we measured and according to Figure 12, it better matches the group speed if the mass loading applies. We admit that flexural waves might exist, but they would travel in the ice unnoticed from the footage. We will alter the discussion to reflect this possibility, as well as exploring the effect of lower values of $Y^*$ on the results.

Comment

Line 371–379. The focus of my concern signalled in the previous item-30 targets the issue of whether the sea ice is behaving as a mass loading medium or as a flexible plate, as the latter depends strongly on the value of the flexural rigidity $Y^*h^3/12(1-\nu^2)$, primarily expressed via thickness h but also the effective modulus $Y^*$. If h or $Y^*$ are over estimated then the dispersion relation will be incorrect. The somewhat uninspiring publication Squire, V.A. A comparison of the mass-loading and elastic plate models of an ice field. Cold Regions Science and Technology 21:219–229 (1993) points out why this is so, namely that the mass-loading dispersion relation is just the flexural plate one with zero flexural rigidity. The authors should take a look at Fig. 7 of that paper, which shows how wavelength is affected by a change of flexural rigidity at a fixed thickness, i.e. a change of $Y^*$. Getting $Y^*$ or h right is crucial. My question is how does it affect the very strong argument for the mass-loading model made in lines 376–379, where the authors state categorically 'It shows quite clearly that at this frequency, waves that are responsible for the break-up and that are visible from the UAV propagate following the mass loading dispersion relation.

Answer

We understand your criticism regarding the conclusions made relative to the dispersion relation constraining wave propagation into the ice, such that waves obeying to mass loading are in fact flexural waves propagating in a low flexural rigidity medium. We will discuss this with more care about underlying concepts. please refer to the previous comment.

Comment

Line 381–382. I note here and on line 141 that the NBB floe was 540 m wide but I cannot find a statement about the GSL ice cover. Admittedly, it is rather late to be asking this question but is there any possibility that the far side of the NBB ice floe is affecting the breakup by creating a standing oscillation under the floe? (A similar question can be asked for the GSL experiment but I don't have the details and the question may be irrelevant.

Answer

We think that since breakup happens fast, there is no time for the development of a standing wave in the ice floe. From the video footage, it is clear that the incident wave crests propagating in the ice are responsible for the breakup.

Comment

Line 390. The authors write 'Based on the previous discussion, we use the mass loading dispersion relation when the wave propagates in unfractured ice.' So, given my comment 31 above in regard to Fig. 7 of Squire (1993) can the authors be sure that it is actually the mass-loading model or could it be a flexible thin plate with a lower value of $Y^*$ or h? For example, page 225 of Squire (1993) states 'Fig. 7 shows that the choice of $Y^*$ is critical in determining the relative magnitudes of the wave numbers in ice and water which, since the elastic plate analogy is a parameterization, suggests caution.' Because the mass-loading model has been dismissed so many times in the wave-ice interaction literature, even for propagation in frazil ice, I am afraid I favor a reduced flexural rigidity hypothesis articulated via $Y^*$ or h.

Answer

Please refer to our previous response.

**Comment**

Line 391. 'It is unclear whether the same relation applies to fragmented ice or if it rather follows the deep water relation'. It is obvious that fractured ice will not have the same dispersion relation as a continuous ice plate if the dispersion relation is based upon a Kirchhoff-Love plate (Euler-Bernoulli beam), but now one enters the murky realm of parameterization. Essentially one could envision a reduction in $Y^*$ when the ice breaks up. Or one could imagine a slightly more extreme version where after breakup $Y^* \to 0$, i.e. the medium becomes a mass-loading medium. If the ice floe starts out behaving as a mass-loading medium then physically I guess one could argue that the masses somehow change ... but they don't as far as I know!

**Answer**

What we meant is that we weren't able to assess it in our observations. We will make this clearer. We agree with the reviewer how different models of ice behaves differently. What we try to assess here is in essence what models would apply best to the ice we are observing.
We suggest changing the sentence to something like: "We were not able to measure waves in fragmented ice and thus it was not possible to characterize the dispersion relation in this part of the ice cover."

**Comment**

Line 395. I recommend adding 'open-water' as it is slightly unclear as it stands; so 'The largest open-water wave was slightly less than one meter high.'

**Answer**

Corrected.

**Comment**

Line 414. I am bothered by 'Even though the ice is thicker in the NBB experiment, the attenuation is almost one order of magnitude weaker than the attenuation in the GSL.' This actually suggests to me that Eq. (17) is producing the wrong ab, i.e. either $Y^*$ or h is incorrect (which would also produce an inaccurate k2).

**Answer**

This is our intention to take into account uncertainties on all parameters before concluding anything. Based on your recommendations we will consider a greater range for $Y^*$ in our calculation of the figures 12 and 13 in order to bring a bit of nuance to the idea proposed. Thank you for underlining the sensitivity of our calculations on $Y^*$ and $h$. Anyhow, we are pretty confident that our measurement of breakup speed and in-ice wave phase speed are valuable so that our results stress how observations are critical to better constrain break-up models.

**Comment**

Line 420. There is no doubt that inelastic effects could cause a difference in attenuation between the two experiments because, as I understand it, the physical properties of the sea ice and hence its material properties were different between the two experiments. (There is no need to say anelastic, as noted elsewhere, as inelasticity covers a multitude of sins.) However, given what I have said above and the lack of amplitude measurement in the ice cover, perhaps any statements about inelastic constitutive laws are best avoided.

Answer
Thank you for pointing out that detail, we will rewrite the text accordingly. For sure it would be amazing to have buoys measuring the curvature of the ice in future experiments.

**Technical corrections**

Comment
The authors declare that they provide the first in situ observations of floe size distributions (FSD) resulting from wave-induced sea ice breakup. Being a pedant, I would aver that the statement is misleading as there are numerous observations of the FSD arising as a result of wave-induced sea ice breakup but that the observations have not segued into any interpretation which improves our understanding of the process. What the authors actually mean is that they are the first to have computed the FSD from observational data of wave-induced breakup. While I actually also find this statement hard to believe, I shall take the authors at their word and assume that they have perused Russian, Japanese and Chinese, as well as North American and European sea ice corpora. However, I also caution that if they haven't scrutinized the international scientific literature carefully, they should avoid making such a strong statement to avoid a rebuttal.

Answer
The abstract has been changed to :
"We provide high resolution in situ observations of wave-induced sea ice breakup in the natural environment. In order to obtain such data, an unmanned aerial vehicle was deployed from the Canadian Coast Guard Ship *Amundsen* as it sailed in the vicinity of large ice floes in Baffin Bay and in the St. Lawrence Estuary, Canada. The footage obtained from these experiment allows for the analysis of both the resulting floe size distribution (FSD) and the temporal evolution of the breakup. When expressed as probability density functions weighted by the floe area, FSDs exhibit a modal shape thus indicating that there is a preferential size associated to wave-induced break-up. Both FSDs are compared to a flexural rigidity length scale, which depends on ice properties, and with the wavelength scale. This comparison tends to show that the maximal distance between cracks is preferentially dictated by sea ice thickness and rigidity rather than by the wavelength. Temporal analysis of one fracture event shows that the break-up advances almost as fast as the wave energy and that waves responsible for the break-up propagate following the mass loading dispersion relation. Moreover, our experiments show that thicker ice can attenuate wave less than thinner ice. This novel dataset and thus provides key information on the wave-induced ice break-up, that has the potential to strengthen theoretical aspects of wave-ice interactions and their implementation in models."

**Comment**

There are various instances of poor English grammar, e.g. singular words that should be plural, poor syntax where the subject is missing, weak or confusing sentence structure, a missing (adjective or adverb) article 'the', etc. In lines 116–118, for instance, '... while still allowing to study break-up in the natural environment. Such a setup also allows to have no constraint on the location of deployments and to search for the right sea ice to break.' Or 'The error on its vertical and horizontal position are respectively of 0.5 and 1.5 m.' Blunders are scattered throughout the text and need to be copy-edited out by somebody as there are quite a few indiscretions, assuming that The Cryosphere expects sound English prose.

**Answer**

We will take a closer look at the grammar when editing the manuscript to be submitted in its final version.

**Comment**

The authors use the word 'further' consistently. In many cases, although not all, they actually mean 'farther'. Such occurrences should be corrected.

**Answer**

See comment above.

**Comment**

There is a mixture of US and English spelling, e.g. 'traveled' yet 'modelled'.

**Answer**

See comment above.

**Comment**

Line 65. 'They rather let it evolve ...' should be 'Rather they let it evolve ...'

**Answer**

Corrected.

**Comment**

Figure 4 caption should read Matlab not Maltab.

**Answer**

Corrected.

**Comment**

Line 154. 'consists in a series of steps' should be 'consists of a series of steps' or 'proceeds in a series of steps'.

**Answer**

Corrected.

**Comment**

Line 161–162. 'We refer the reader to (Zhang and Skjetne, 2018)' should be cited as 'Zhang and Skjetne (2018)'.

**Answer**

Corrected.

**Comment**

Line 220. 'let's compute' is a little informal.

**Answer**

Line 220 has been changed to :
"To characterize the FSD, the floe size distribution (NFSD) is computed using the frequency of observation as it is done in most studies ..."

**Comment**

Line 278. The brackets in the citation are of the wrong type, i.e. it should be Fox and Squire (1991).

**Answer**

Corrected.

**Comment**

Line 310. '... lies on the same mathematical premises?' is poor English. Replace with '... based upon the same mathematical premises?'

**Answer**

Corrected.

---

## Author Response (AR2)

**Review of "Aerial observations of sea ice break-up by ship waves" by Dumas-Lefebvre & Dumont (2021)**

**General appreciation**

Comment

> This is a challenging and valuable paper to attempt to clarify the sea ice break-up processes and resultant floe size distributions through wave-ice interaction from "artificial" field experiments. By inducing waves with a cruising ship, how waves broke sea ice floes was monitored by a drone. This experiment was conducted in winter and summer, and the results were assessed to improve the understanding of the break-up processes of sea ice and the generation of floe size distribution. The major findings are the importance of the ice rigidity as well as wavelength to determine the maximum floe size, much slower propagation speed of ice break-up compared with the expected wave propagation speed, and the influence of ice thickness on the wave attenuation. They also proposed a new method to represent the floe size distribution.
>
> It is well known that the floe size distribution is one of the key parameters of sea ice and the ice break-up processes due to waves play an important role in its formation. However, due to logistical difficulties, it has been quite difficult to clarify the break-up processes from in-situ observations. To my knowledge, this is the first experiment conducted in the real sea ice area under the idealized conditions. Therefore, basically I support this work and recommend publication. Having said that, I feel some descriptions seem confusing and need to be reconsidered. For me, some conclusions are not necessarily clear. I would appreciate it if the authors address them before publication. If it comes from the lack of my knowledge or my misunderstanding, please forgive me.

Answer Thank you for reviewing our paper. Your comments helped us to increase the quality of our manuscript. Please see below for the answers to your major and specific comments.

**Major comments**

Comment

> In this experiment, ice thickness is one of the key parameters. Therefore, I think the measurement method and accuracy should be described clearly. To my understanding, it was obtained from the ice chart for GSL (L134) and from meter stick (L147-148). Since the description is only brief, I wonder if it is possible to use these data obtained from different sources equally for quantitative assessment. Some additional descriptions to guarantee it would be desirable. Besides, the horizontal scale seems to have been determined only from the FOV of the camera and the flight height of the drone (L127-128). Since the view angle and flight height could contain some ambiguities, it would be ideal to check the horizontal scale from the real scale such as the ship length if possible. It was regrettable that buoy data were not available (L154) because the information of incoming waves is quite important

Answer

The ice thickness and image resolution are indeed key parameters of the experiments that allow to compare observations against theory and to scale the majority of the breakup properties. Unfortunately, as the reviewer points out, we were unable to hop on the ice and measure its thickness at various places from drill holes. We were only able to measure the thickness during the Baffin Bay experiment directly by going out on a zodiac and using a meter stick with a hook at one end at the edge of broken up floe. We gave to this measurement a somewhat large uncertainty (10 cm) in our analyses in order to compensate for various possible sources of errors.

In the GSL experiment, the thickness of the unbroken ice floe was assessed visually, looking at the ice freeboard while standing on the ship's lower deck 3 m above the ice. The value was also given a large 10-cm uncertainty for analyses. We also put that information in the context of the Canadian Ice Service ice chart which is also coherent with what we saw.

We modified the text of section 2.1 and 2.2 as well as Figures 1 and 2 to describe in greater detail prevailing ice conditions and how we estimated ice thickness.

Like for the error on altitude, the error on horizontal dimensions due to the uncertainty on the camera field of view is very low. We appended the error assessment at L145 with details on the impact of the error on the FOV, which is conservatively estimated to $\pm 0.5°$. It leads to an error of 0.03 cm px$^{-1}$ (0.9%), which is still two orders of magnitude smaller than the pixel size (3.1 cm). The total error associated with both paramters is 0.04 cm px$^{-1}$.

Comment

In section 3.1 they described "the minor axis length is the chosen floe length scale as it represents the characteristic break-up length scale" (L186-187). If they mean the ice break-up due to major propagating waves, the orientation of the minor axis should be aligned in the similar direction. But as far as looking at Fig. 4, the directions of the minor axis are variable. I guess various factors affected the ice break-up processes. I want the authors to explain what they meant by the characteristic break-up scale. Besides, I would like to know how the major axis length was determined and the relationship between the two parameters. I think this is important because the selection of floe size is relevant to the Area-based floe size distribution in Fig. 10.

Answer

Using the minor axis of ellipses fitted to ice floes is convenient to rapidly assess the horizontal scale that is relevant to flexural break-up, and other studies have used that metric (see for e.g. Herman et al. 2018). When waves break-up the ice, sometimes floes are very anisotropic, and the smallest dimension occurs along the wave propagation direction. For such elongated floes, the fitted ellipse is very well aligned with the main axes. Of course, when floes are nearly isotropic, the fitted ellipse may not be aligned with the wave directional axes, but since the two length scales are similar, the number that is computed is still representative. Of course, the shape of floes produced during a wave-induced break-up do not comply perfectly with ideal behavior. However, the advantage of this method is that it is fast and that all floes are taken into account in the FSD. Other methods would be possible, but all would have their limitations. Looking at Fig. 4, one sees that floes that deviate the most from the wave direction (minor axis along the wave direction) are nearly isotropic or are smaller than average. In the former case the error on the size is low, and on the latter case these floes do not contribute significantly to the shape of the AFSD.

Comment

Based on the ratio of Cg to Cp (Fig. 11), they inferred that the break-up speed was controlled by the mass loading effect. Although it might be true, I am a bit skeptical about the idea. This approach would apply to the waves propagating in the fractured ice area. But I consider the ice break-up mechanism would be involved to determine the break-up speed, and the situation is not necessarily the same as the mass loading effect. Some additional effect such as ice fatigue, as discussed by Langhorne et al.(1998), or the heterogeneous properties of sea ice should be involved. If they consider the break-up mechanism is not so important, please add some explanation.

Answer

In this study (GSL experiment) we measured three things: 1) the wavelength in unbroken ice, 2) the wave period in broken and unbroken ice and 3) the speed at which break-up occur in the direction of the wave, that we call $c_b$. We do not measure the group speed. The phase speed is defined as the wavelength (distance between two crests/troughs) divided by the period (time it takes for one crest/trough to travel at the position of the previous one). This is the speed we can estimate directly. What we try to do is to determine how does $c_b$ relate to the wave group propagation. Figure 11 shows that $c_b$ is slower than the group speed of open water waves and that mas-loaded wave speed is within the uncertainties of our measurements. After relflection, your comment made us realise that the breakup speed will be shifted from the group speed, either negatively due to attenuation or positively due to ice fatigue. The identification of the dispersion relation of wave propagating in unbroken ice can be made with the phase speed and that is why we added a subfigure to Figure 11. Section 5.2 was modified to clarify this interpretation, which in our opinion greatly improves the manuscript.

**Comment**

I think the [conclusion] can be more concise, focusing on the essence of the new findings. Regarding the novel way of computing the FSD, it might be better to add some general explanations about how effectively it represents the physical properties of FSD compared with the traditional way. I am wondering why they focus on $d_{max}$ although they use $d_{min}$ for drawing the AFSD.

**Answer**

We added a better description of the advantages of the AFSD over the NFSD and modified the text in the conclusion in order for it to be more concise.
We focus on $d_{\max}$ because the theoretical frameworks prescribe values of $d_{\max}$. We would have done otherwise if there were theories for the modal size or $d_{\min}$, but there is unfortunately none that focuses on the position of the critical strain. This would be helpful for parametrizing the FSD from sea ice and wave properties in larger scale models.
We do not use $d_{\min}$ for drawing the AFSD. I guess you refer to $x*$ which is close to the minimal sizes in the GSL and NBB AFSDs. We think that the value of $x*$ being close to the minimal size in both experiment exhibits the fact that this semi-static framework is not well suited for breakup since it should represent the maximal size and it does not.

**Specific comments**

**Comment**

* (Abstract, L11) ".. thicker ice can attenuate wave less than thinner ice." Is that true? To my understanding, in the thicker ice situation of NBB, the wave attenuation was less because of the less ice rigidity caused by more brine volume fraction.

**Answer**

It is. The ice was thicker in the NBB and the extent of breakup was larger. This highlights the fact that brine volume, and thus Young's modulus, are important values in determining if sea ice is going to be broken by waves or not.

**Comment**

*(Introduction) I think the background of this study was well researched. But if you agree, please consider adding the following papers for the observational studies that directly relate the FSD (L38): Kohout, A.L., et al. (2016): In situ observations of wave-induced sea ice breakup. Deep-Sea Research II, 131, 22-27. And for the analysis of the resulting FSD and its possible connection to sea ice flexural rigidity (L86): Toyota et al. (2011): Size distribution and shape properties of relatively small sea-ice floes in the Antarctic marginal ice zone in late winter. Deep-Sea Research II, 58, 1182-1193.

Answer

Thank you for your suggestions. After further examination of Kohout et al. (2016), there is indeed a thorough quantification of wave conditions during each breakup event and an attempt to predict the breakup extent with a strain model. Unfortunately, there is no FSD extracted from their dataset. They do mention that the "[evenly spaced cracks suggest] a relationship between wave induced ice breakup and floe size distribution" but do not proceed to further analysis. We will thus add add this reference at line 85 since this portion of the paper mentions the other wave-induced ice breakup already made.
We will add Toyota et al. (2011) at line 86.

Comment

*(Introduction, L55) Please add the definition of "WIMs".

Answer

Corrected, thank you.

Comment

*(Section 3.2, L192) "The wave phase speed is then obtained.." I am wondering that judging from the estimation method, this might correspond to the wave group velocity (Cg). It seems that they treated this value as the group velocity in section 5.2, didn't they? To my understanding, the cruising ship generated the fixed frequency, which induced the waves with various two-dimensional wave numbers satisfying the dispersion relation. The observed large wave amplitude evolution corresponds to the maximum group velocity produced by them.

Answer

As discussed in a response to a previous comment, what we measure is the phase speed, defined as $c_p = \omega/k = \lambda/T$. We did not measure the group speed, but rather inferred it using existing theories.
Note that the phase speed of ship waves is also given by $c_p = U\cos(\theta)$, which is consistent with our independent and more direct measurement in ice, within uncertainty limits.

Comment

*(Section 4.2, L228) ".. the minor axis d since.." I am wondering if the orientation of the minor axis might be important as well. I want to know how the length of the major axis was determined. Is that because of the heterogeneous properties of sea ice or any other reasons?

The orientation of the minor axis, which may have a slight offset relative to the *true* inter-crack distance, could contribute to the spread of the FSD. But, since the orientation distribution of the floes (result not shown) is narrow, it is likely not a major contributor to the spread of the FSD as the heterogeneity of the ice properties might be.

The length of the major axis is determined by MATLAB's Image Processing Toolbox and has the following definition : "Length (in pixels) of the major axis of the ellipse that has the same normalized second central moments as the region, returned as a scalar."

To support our choice of metric for the floe size, we added Herman et al. (2018) as reference who did the same choice for the same reasons.

*(Section 5.1, L270-271) "The shape of the AFSD also highlights the fact that this process alone does not explain the power law distribution..." This is interesting. To show it clearly, how about displaying the (traditional) cumulative FSD directly, if you agree?

Thank you for your suggestion. Nonetheless, we consider that the non-cumulative FSD used in the paper displays efficiently the fact that waves alone do not generated a power-law FSD and allows for an easier translation into models which already use binned distribution of ice thickness.

*(L277) "propose" should be "proposed".

Corrected, thank you.

*(Section 5.1, L319-320) "Toyota et al. (2011) to consider that.." Since they introduced this idea to explain the threshold of two regimes, I feel that their concept is not inconsistent with the authors' idea described in this paper.

Even if it was used by them for the threshold between two regimes, they still based their rationale on the fact $x^*$ should represent a minimal fracture distance which turned out to be a wrong interpretation of Mellor (1983). That is why we mention it.

*(Section 5.1, L350-354) Equation 14 & 15 According to the definition, it seems that $\beta$ corresponds to the ratio of x* to wavelength. Please add some description about the physical meaning of $\beta$, which would facilitate the readers' understanding.

Answer

It is indeed the ratio $(x^*/\lambda)^4$ and comes from the dimensional analysis of the boundary condition on $\eta$, eq. 1.3 in Tkacheva (2001). We added the following text between eq. 14 & 15 to make it clearer to the readers : "This quantity arises from the dimensional analysis of the the boundary conditions considered for solving the velocity potential of a plane wave being diffracted by a plate."

Comment

*(Section 5.1, L367) "an" should be "and".

Answer

Corrected, thank you.

Comment

*(Section 5.2, L406-408) "this result rather suggest that. . ." This is an interesting result and I agree.

Answer

Great !